# Neural Synchrony Between Socially Interacting Language Models

**Zhining Zhang[1], Wentao Zhu[2], Chi Han[3], Yizhou Wang[1], Heng Ji[3]**
[1] Peking University    [2] Eastern Institute of Technology, Ningbo
[3] University of Illinois Urbana-Champaign
zzn_nzz@stu.pku.edu.cn

## Abstract

Neuroscience has uncovered a fundamental mechanism of our social nature: human brain activity becomes synchronized with others in many social contexts involving interaction. Traditionally, social minds have been regarded as an exclusive property of living beings. Although large language models (LLMs) are widely accepted as powerful approximations of human behavior, with multi-LLM system being extensively explored to enhance their capabilities, it remains controversial whether they can be meaningfully compared to human social minds. In this work, we explore neural synchrony between socially interacting LLMs as an empirical evidence for this debate. Specifically, we introduce neural synchrony during social simulations as a novel proxy for analyzing the sociality of LLMs at the representational level. Through carefully designed experiments, we demonstrate that it reliably reflects both social engagement and temporal alignment in their interactions. Our findings indicate that neural synchrony between LLMs is strongly correlated with their social performance, highlighting an important link between neural synchrony and the social behaviors of LLMs. Our work offers a new perspective to examine the "social minds" of LLMs, highlighting surprising parallels in the internal dynamics that underlie human and LLM social interaction.[1]

## 1   Introduction

Humans are fundamentally social: when people talk, collaborate, or even simply share attention, their brain activities begin to synchronize (Dumas et al., 2010; Hasson et al., 2012; Kawasaki et al., 2013). This phenomenon, known as *inter-brain synchrony* (IBS), is not merely a byproduct of shared sensory input. Instead, it predicts and facilitates crucial aspects of social dynamics, including coordination, cooperation, and mutual understanding (Mu et al., 2016; Dikker et al., 2017; Bevilacqua et al., 2019; Davidesco et al., 2023). This suggests that human social cognition is supported by physical mechanisms that allow human minds to align with each other.

While Large Language Models (LLMs) have shown remarkable approximation of human social interaction (Bianchi et al., 2024; Piatti et al., 2024; Breum et al., 2024), with multi-LLM systems being explored to enhance their capabilities (Zhang et al., 2023; Du et al., 2023), whether they truly resemble human social minds remains unknown (Shapira et al., 2023; Zhou et al., 2023b; Street, 2024). In this work, we study interacting LLMs at the representation level, and demonstrate that LLMs exhibit an analogue of neural synchrony during social interaction, not referring to biological activity, but to the synchrony of their internal representations. This provides the first empirical evidence that they also exhibit an internal mechanism for supporting social interaction.

We measure emergent neural synchrony as the alignment between the representations of interacting LLMs, which occurs only when (1) the LLMs are *socially engaging* and (2) their representations are *temporally proximal*, paralleling the dynamics observed in human IBS. This alignment serves as a salient indicator that the LLMs' representations can distinguish real-time social interactions from mere contextual similarity, thus exhibiting a form of social awareness. As illustrated in Figure 1, we simulate social interactions and collect the neural representations between socially interacting LLMs. To measure their neural synchrony, we train affine transformations that predict one LLM's

---

[1]The code is available at https://github.com/zzn-nzz/LM_neural_synchrony.

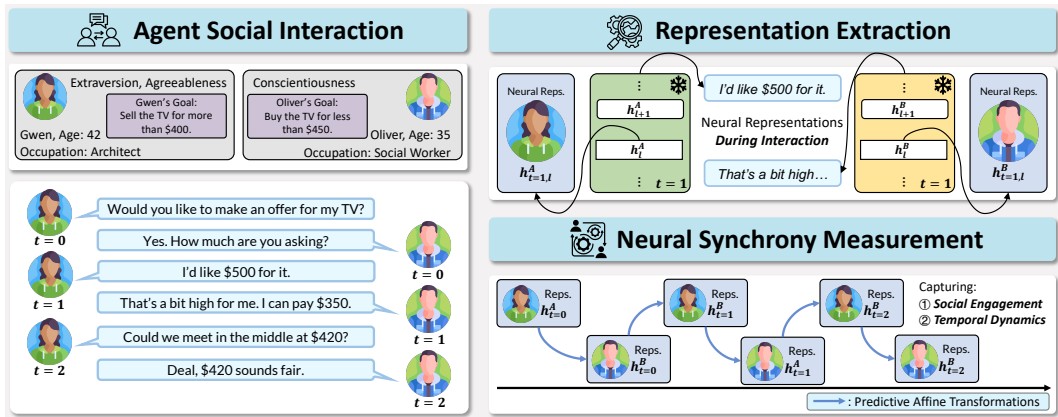

Figure 1: An illustration of our analysis framework. (1) Two LLM agents engage in a social interaction, generating responses conditioned on their backgrounds, goals, and shared history. (2) Hidden representations are extracted at each turn from both LLMs. (3) Neural synchrony is measured by learning affine mappings to predict one agent's future representations from the other's current ones.

future representations from the other's current representations. We further quantify the synchrony by defining a metric $SyncR^2$ over the prediction performance of these transformations.

We show that the proposed measurement, $SyncR^2$, reflects *social engagement* and *temporal proximity* through carefully designed experiments covering various LLMs. First, interacting LLMs show significantly higher $SyncR^2$ than non-interacting controls, showing an emergent synchrony only when the two agents are both engaged in genuine interaction. Second, $SyncR^2$ depends on temporal alignment and declines sharply under a temporal lag, showing that the measure reflects real-time representational dynamics between LLMs instead of static representation similarity. Finally, We demonstrate a strong correlation between **representational-level** neural synchrony in LLMs and their **behavioral-level** social performance. Based on experiments with 21 model pairs spanning diverse families and sizes, the results are statistically significant. Importantly, the correlation remains robust after controlling for potential confounding factors, i.e., instruction following and long-context reasoning abilities of models.

Our main contributions include: (1) We are the first to study the representation dynamics between socially interacting LLMs. We introduce a novel framework for measuring neural synchrony between socially interacting LLMs, based on predictive transformations between representations. (2) Through carefully designed control experiments, we demonstrate that measured neural synchrony reflects both social engagement and temporal alignment. (3) We show that neural synchrony between LLMs is strongly correlated with their behavioral-level social performance across diverse model families. Together, these contributions provide the first empirical evidence that socially interacting LLMs exhibit an internal mechanism analogous to human inter-brain synchrony, opening a new lens for studying the "social minds" of LLMs.

## 2 RELATED WORK

### 2.1 INTER-BRAIN SYNCHRONY

The discovery of *inter-brain synchrony* (IBS) between interacting people represents a pivotal advancement in understanding the neural mechanisms underlying human sociality. IBS refers to the temporal alignment of neural activity between individuals during social interaction, typically measured using hyperscanning techniques (Dumas et al., 2010; Astolfi et al., 2010; Nam et al., 2020; Lu et al., 2023). IBS has been observed in socially engaging contexts (Kawasaki et al., 2013; Koul et al., 2023) and even during cooperative tasks without direct sensory exchange (Lu et al., 2023). The strength of IBS varies with social attributes such as interpersonal closeness and personality traits (Kinreich et al., 2017; Bevilacqua et al., 2019). More importantly, IBS is not just an epiphenomenon of interaction but also a predictor of meaningful social and cognitive outcomes. Higher levels of

IBS have been linked to greater cooperation rates (Hu et al., 2018), enhanced learning effectiveness (Bevilacqua et al., 2019), and improved team performance (Reinero et al., 2021).

## 2.2 BRAIN-LLM SIMILARITIES

Recent works have highlighted parallels between LLM and human brain representations. Aw et al. (2023) find that instruction fine-tuning improves brain alignment; Mischler et al. (2024) demonstrate that cortical hierarchies and LLMs converge on contextual *textual* feature extraction. Doerig et al. (2025) show that embeddings from LLMs align with high-level *visual* brain representations, suggesting shared contextual coding across modalities. In the *social* domain, Jamali et al. (2023) report that LLM embeddings selectively respond to true- and false-belief tasks, akin to single neurons in the human dmPFC. These studies uncover shared underlying connections between humans and LLMs, but focus on representations of single models with static inputs. In contrast, our work examines whether such parallels extend to multi-round, socially engaged interactions between LLMs.

## 2.3 "SOCIAL MINDS" OF LLMS

A growing body of work aims at examining whether LLMs possess true "social minds". One line of research develops social simulation environments to evaluate LLMs in various social tasks (Bianchi et al., 2024; Piatti et al., 2024; Breum et al., 2024; Zhou et al., 2023c), focusing on the social capability of LLMs. Another line evaluates Theory of Mind (ToM) reasoning in LLMs (Zhou et al., 2023b; Shapira et al., 2023; Street, 2024), which shows that state-of-the-art LLMs still does not possess human-level ToM (Gandhi et al., 2023; Kim et al., 2023; Jin et al., 2024; Chen et al., 2024).

While these studies provide valuable insights, they primarily assess social ability at the behavioral level. By contrast, Zhu et al. (2024) and Bortoletto et al. (2024) examine the internal Theory of Mind representations of LLMs. They find that certain attention heads differentiate between self and other beliefs, akin to Theory of Mind-related brain regions in humans (Siegal & Varley, 2002; Carrington & Bailey, 2009). However, their analyses remain limited to single-model social reasoning from a third-person perspective, rather than multi-round multi-model social interaction with genuine engagement. Our work complements these directions by moving beyond behavioral evaluations and static reasoning analyses and investigating representation-level neural synchrony between socially interacting LLMs, thereby offering a novel perspective on examining the "social minds" of LLMs.

## 3 MEASURING NEURAL SYNCHRONY BETWEEN LLMS

In humans, inter-brain synchrony is measured from continuous neural signals like EEG, where phase-locking values or total interdependence across time captures synchrony (Dumas et al., 2010; Dikker et al., 2017; Nam et al., 2020). LLMs instead produce discrete representations per interaction turn, limiting standard neuroscience methods. To measure the social temporal dynamics of LLM interactions, we propose to train predictive affine transformations between LLMs. Our approach is motivated by the hypothesis that, during social interaction, LLMs not only approximate behaviors based on their own agent profiles, but also actively engage by reasoning about their partner's emotions, intentions, and how the interaction may unfold. If so, then the internal representations of one LLM should contain information predictive of the other's representations. Therefore, We train predictive affine transformations and interpret their predictability to measure neural synchrony.

### 3.1 DATA COLLECTION

**Environment.** We employ SOTOPIA (Zhou et al., 2023c) to simulate social interactions. SOTOPIA is an environment where agents role-play characters to achieve social goals in complex scenarios, including coordination, competition, negotiation, persuasion, and everyday social encounters. The task space is large and diverse, constructed from a wide range of scenarios, characters, and relationships to create realistic settings. Within this environment, each agent has its own background, including personalities, public information, and even secrets. We include examples of scenarios, agent profiles, and interactions in SOTOPIA in Appendix A.

**Interaction simulation and representation extraction.** A simulated interaction includes multiple turns. At each turn, an LLM is given a prompt that includes the agent's profiles, goals and inter-

action history. Note that two agents in the same interaction have distinct profiles and their goals are not accessible to each other. When an LLM generates response, we extract the hidden states at the final token position of the prompt input, since it integrates information from all previous tokens. Throughout this work, we refer to these hidden states as the model's internal *representations*. Formally, for a dialogue with $T$ turns, for each LLM backbone $M \in \{A, B\}$, at turn $t$ we obtain

$$\boldsymbol{h}_t^{(M)} \in \mathbb{R}^{L_M \times D_M}, \quad t = 1, \dots, T,$$

where $L_M$ and $D_M$ represent the number of layers and layer dimensions of $M$, respectively.

## 3.2 MEASURING NEURAL SYNCHRONY BETWEEN LLMS

To assess the social temporal alignment between socially interacting LLMs, we ask whether the representations of agent $A$ can be linearly transformed to predict those of agent $B$ at the next turn. This is similar to representation alignment measurements (Huh et al., 2024; Zhang et al., 2025), but incorporates a temporal correspondence between measured features.

**Dataset construction.** We gather representations from the LLM agents in simulated interactions over diverse scenarios to construct the dataset for measuring neural synchrony. At each turn $t$, we collect the hidden representations $\boldsymbol{h}_{t,l_A}^{(A)}$ from agent $A$ at layer $l_A$ and $\boldsymbol{h}_{t,l_B}^{(B)}$ from agent $B$ at layer $l_B$. These temporally aligned pairs are used to construct the dataset (Experimental group):

$$\mathcal{D}_{l_A \to l_B}^{A \to B} = \left\{ \left( \boldsymbol{h}_{t,l_A}^{(A)}, \boldsymbol{h}_{t,l_B}^{(B)} \right) \, \Big| \, t = 1, \dots, T \right\}.$$

**Learning affine transformations.** To learn the predictive mapping between two layers' representations, we apply ridge regression with intercept, taking $\boldsymbol{X}$ as the input and $\boldsymbol{Y}$ as the target representations in dataset $\mathcal{D}$:

$$\hat{\boldsymbol{W}}, \hat{\boldsymbol{b}} = \underset{\boldsymbol{W}, \boldsymbol{b}}{\arg\min} \|\boldsymbol{Y} - \boldsymbol{X}\boldsymbol{W} - \mathbf{1}\boldsymbol{b}\|_F^2 + \lambda\|\boldsymbol{W}\|_F^2$$

**Measuring neural synchrony.** For any model pair $(A, B)$, we train and evaluate the learned affine transformation on $\mathcal{D}_{l_A \to l_B}^{A \to B}$ for each layer pair $(l_A, l_B)$ and compute $R_{test}^2(l_A \to l_B)$, the coefficient of determination, which indicates the explained variance of the target representation. Figure 2 shows example synchrony heatmaps showing $R_{test}^2(l_A \to l_B)$ for all $(l_A, l_B)$ between two models.

As the best-predicted target layer varies across different source layers, we summarize these predictive strengths by further aggregating over all layers in $B$ as

$$r_A^\star(l_A) \;=\; \max_{l_B \in \{1, \dots, L_B\}} R_{test}^2(l_A \to l_B), \qquad \tilde{r}_A(l_A) \;=\; \max\{0, \, r_A^\star(l_A)\}.$$

Then, we define the neural synchrony score of model pair $(A, B)$ as

$$SyncR^2(A \to B) = \frac{1}{L_A} \sum_{l_A=1}^{L_A} \tilde{r}_A(l_A), \quad SyncR^2(A, B) = \frac{1}{2}(SyncR^2(A \to B) + SyncR^2(B \to A))$$

*Intuition.* For each layer in $A$, we take its best predictive match in $B$. We also clamp negative $R^2$ values to zero, since a negative $R^2$ indicates that the learned mapping performs worse than simply predicting the mean, which we interpret as the absence of synchrony. Clamping to zero ensures that only genuine predictive relationships contribute to the score. The final value is the average of these best-case predictions across all layers of $A$, reflecting how well $A$'s representations can anticipate those of $B$ during interaction. We then compute $SyncR^2 \, (B \to A)$ analogously (with slightly different notation; see Appendix B for details), and average the two directions to obtain a bidirectional measure of neural synchrony. The robustness of the design choices of best predictive match and clipping negative $R^2$ is evaluated through additional sensitivity analyses, which are provided in Appendix C.

## 3.3 SOCIAL ENGAGEMENT AND TEMPORAL PROXIMITY

To examine whether $SyncR^2$ truly captures neural dynamics of interaction rather than spurious correlations (e.g., from shared context or static representational similarity), we validate it under several rigorously controlled conditions. Our analysis centers on two key criteria of neural synchrony: *social engagement* and *temporal proximity*.

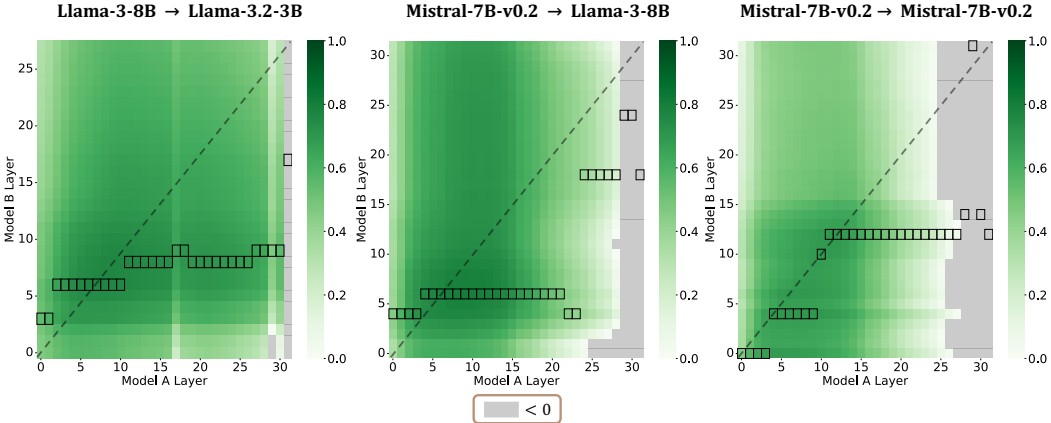

Figure 2: Example synchrony heatmaps of $A$ predicting $B$. The horizontal and vertical axes denote source layers $l_A$ and target layers $l_B$, respectively, with each cell showing the test-set $R^2$ score for the corresponding layer pair. Gray cells indicate negative test-set $R^2$ values. For each $l_A$, the best-predicting $l_B$ is highlighted with a black box. Model names are abbreviated by omitting the suffix "Instruct" due to space constraints. Note that `Llama-3.2-3B-Instruct` has 28 layers, whereas the other models have 32 layers.

**Control group 1 (w/o social engagement).** If synchrony arises from genuine interaction, $SyncR^2$ should decrease when one LLM only passively consumes the dialogue, without the motivation for role-playing and generating response. Therefore, we collect representations from an alternative run where we introduce a "passive" agent who only reads the interaction. It does not receive instructions about generation nor generate responses, but only consumes the same background and interaction history from the normal run. We then measure whether such a passive agent exhibits the same degree of neural synchrony with its partner as interactive agents do:

$$\mathcal{D}_{l_A \to l_B}^{\text{passive}, A \to B} = \left\{ \left( \boldsymbol{h}_{t,l_A}^{(A,\text{read})}, \ \boldsymbol{h}_{t,l_B}^{(B)} \right) \ \Big| \ t = 1, \dots, T \right\}.$$

**Control group 2 (w/o temporal proximity).** Synchrony should capture short-term social dynamics emerging across turns, rather than arbitrary correlations at long lags. We aim to exclude similarities introduced by the shared background profiles, or by representational structures of the interacting LLMs. To test the necessity of temporal proximity, we pair each source representation with a future target representation $k$ turns ahead ($k \geq 1$):

$$\mathcal{D}_{l_A \to l_B}^{\text{lag-}k, A \to B} = \left\{ \left( \boldsymbol{h}_{t,l_A}^{(A)}, \ \boldsymbol{h}_{t+k,l_B}^{(B)} \right) \ \Big| \ t = 1, \dots, T - k \right\}.$$

Together with the experimental group in Section 3.2, these control settings allow us to examine whether measured neural synchrony truly arises from genuine interaction (experimental), or instead reflects passive co-exposure to shared context without active participation (control 1), or simple representation similarity that do not rely on temporal alignment (control 2).

## 4 EXPERIMENTS

We first describe the basic experimental setup in Section 4.1. Then, in Section 4.2, we examine whether the proposed measurement captures the social temporal dynamics of interaction. Finally, in Section 4.3, we analyze how agent relationships shape neural synchrony.

### 4.1 EXPERIMENTAL SETUP

**Models.** We conduct experiments using 6 open-source models: `Mistral-7B-Instruct-v0.1`, `Mistral-7B-Instruct-v0.2` and `Mistral-7B-Instruct-v0.3` (Jiang et al., 2023); `Llama-2-7B-Chat` (Touvron et al., 2023), `Llama-3-8B-Instruct` and `Llama-3.2`

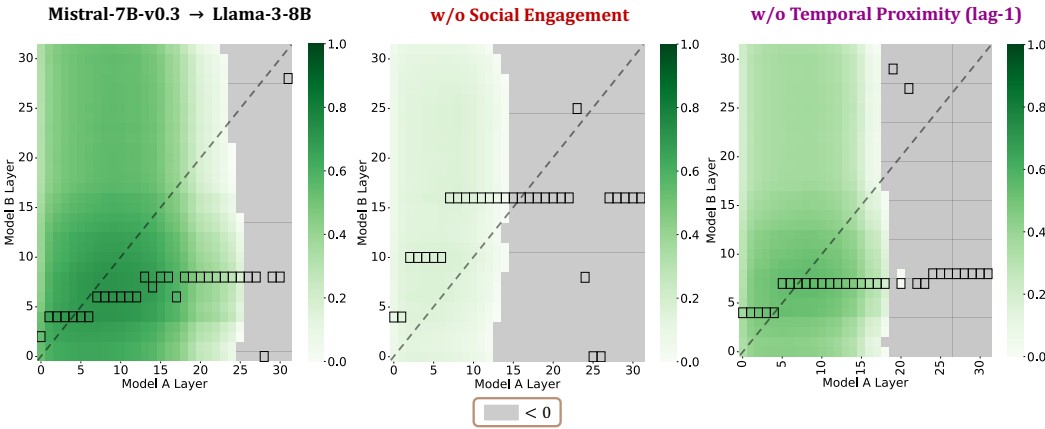

Figure 3: Examining synchrony of `Mistral-7B-Instruct-0.3` predicting `Llama-3-8B-Instruct` under different conditions. Left: Experimental group with genuine interaction. Middle: Control group 1 (*w/o social engagement*), where one agent passively consumes the dialogue without actually engaging in the interaction. Right: Control group 2 (*w/o temporal proximity*), where source representations are paired with lagged future representations.

`-3B-Instruct` (Dubey et al., 2024). The *Mistral* family provides a sequence of models with consistent architecture emphasizing improved efficiency and performance. The *Llama* family consists of widely adopted models spanning multiple sizes. Together, these two families provide a diverse testbed for our study. From these six models, we construct 21 distinct model pairs, covering both intra-family pairings (e.g., different *Mistral* versions) and cross-family pairings (*Mistral & Llama*).

**Datasets.** We construct datasets for both the experimental group and the control conditions as described in Section 3.2 and Section 3.3. Concretely, we simulate 450 interaction scenarios under 3 different random seeds in SOTOPIA for the constructed model pairs. At each turn in each scenario, we collect hidden representations from every transformer layer of both agents, yielding paired samples across all $(l_A, l_B)$ layer combinations. For each ordered agent pair $(A, B)$, we build two directional datasets $\mathcal{D}^{A \to B}$ and $\mathcal{D}^{B \to A}$ to measure neural synchrony from both directions. To ensure fair comparisons, we downsample or sample more and fix the number of samples to 6,500 for all model pairs in all experiments. Pilot experiment on effects of sample size is provided in Appendix D.

**Implementation details.** During simulation, we set the sampling temperature to 0.7 for LLM response generation. We cap each scenario at 8 turns, as interactions beyond this point generally become uninformative. Agents may also end the interaction early if they consider their goals achieved. For data collection, each dataset $\mathcal{D}$ is randomly split into training and test sets with a ratio of 8:2. To prevent for potential data leakage, we ensure that representations from the same interaction are not split between the train and test sets (see Appendix E for more discussions). For training, we use a manually implemented ridge regression with regularization factor $\lambda = 0.1$, where the intercept term is not regularized. For evaluating, we compute the coefficient of determination ($R^2$) on the held-out test set with uniform weighting, averaging results over 3 random seeds.

## 4.2 NEURAL SYNCHRONY CAPTURES SOCIAL AND TEMPORAL DYNAMICS OF INTERACTION

We first examine if $SyncR^2$ captures social temporal dynamics by contrasting between experimental group and controls introduced in Section 3.3. As shown in Figure 3, where example heatmaps are contrasted, prediction performance of the affine transformations is substantially lower in the control settings. In particular, many pairs yield negative $R^2$ values, indicating that the learned mapping performs worse than simply predicting the mean. This suggests that no meaningful synchrony exists between the corresponding layer pairs under these controls.

Figure 4 summarizes the effects of the control settings. In the absence of social engagement (control 1; panel a), the average $SyncR^2$ across models in each family is significantly reduced compared to the experimental group, confirming that synchrony does not arise as much when one agent passively

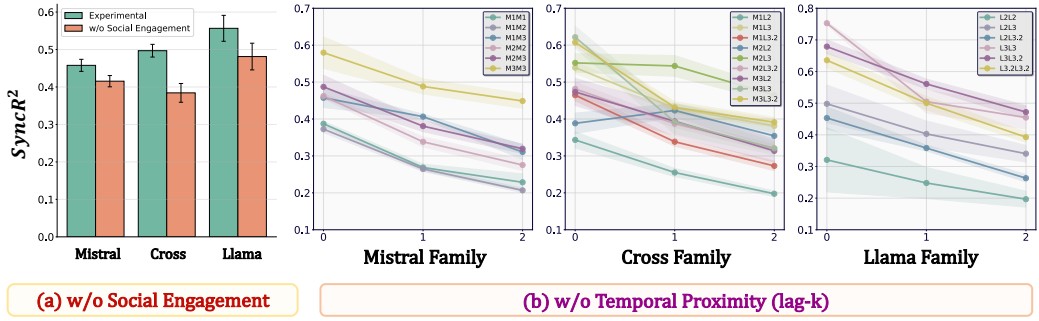

Figure 4: Neural synchrony declines under control settings. **(a)** Control group without social engagement: bar plots show the mean $SyncR^2$ across models within each family. **(b)** Control group without temporal proximity: line plots show $SyncR^2$ for all model pairs within each family as a function of lag $k$ ($k = 0$ is the experimental group). Model names are abbreviated (see Appendix F for the full name correspondence). Error bars denote the standard errors.

consumes the dialogue. In the absence of temporal proximity (control 2; panel b), synchrony collapses when representations are paired across different lags. Together, these results demonstrate that neural synchrony crucially depends on both genuine social engagement and temporal alignment.

### 4.3 Effects of agent relationships

We next investigate how neural synchrony varies under different *agent relationships*, which may shape the extent to which agents are incentivized to align with one another. Neuroscience work has demonstrated that closer human relationships (e.g., couples vs. strangers) are associated with stronger and more efficient inter-brain synchrony during joint tasks (Djalovski et al., 2021). To examine whether analogous effects of agent relationships arise in LLM interactions, we use its defined relationship categories to study how these attributes affect neural synchrony (see Appendix G for implementation details). Figure 5 shows the distribution of $SyncR^2$ across different agent relationship types. The synchrony distribution shifts upward as social closeness increases, suggesting that neural synchrony not only captures social temporal dynamics but is also shaped by the social closeness of the interacting agents.

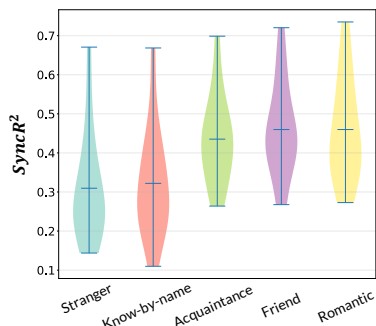

Figure 5: Neural synchrony under different relationship closeness. $SyncR^2$ is aggregated over all model pairs.

## 5 Neural Synchrony as an Indicator of Social Performance

A central question in neuroscience is whether human inter-brain synchrony plays a functional role in enabling successful social interaction. Empirical studies have shown that stronger neural synchrony between humans predicts better coordination, cooperation, and collective performance (Hu et al., 2018; Bevilacqua et al., 2019; Reinero et al., 2021). Motivated by these findings, we ask whether the **representation-level** neural synchrony between LLMs is correlated with their **behavioral-level** social performance.

### 5.1 Experimental setup

We evaluate the social performance of different LLMs in Sotopia, using the automated Sotopia-Eval framework that provides a multi-dimensional evaluation of interactions, including goal completion, impact on relationships, agent believability, and so on (see Appendix A for details). We use gpt-oss-120b (OpenAI et al., 2025) as the interaction evaluator for its cost-effectiveness and close alignment with human evaluation. Quantitative evidence of this alignment is provided in

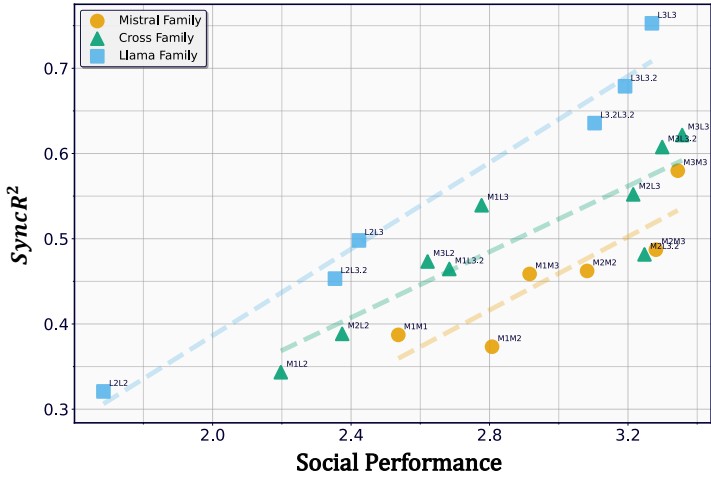

Figure 6: Correlation between neural synchrony and social performance across different family types. The $x$-axis denotes the average social performance of the LLM pair in SOTOPIA across all scenarios, evaluated by `gpt-oss-120b` which aligns closely with human evaluation. The $y$-axis shows the measured synchrony score $SyncR^2$ of the two models. Each point is annotated with abbreviated model names (the full name correspondence is provided in Appendix F). The results show a clear linear relationship between neural synchrony and social performance, where a general trend is that more synchronized models achieve better social performance.

Appendix H. We set the evaluator with temperature $= 0$ and use a fixed seed. For each model pair, we report the average evaluation scores across three simulation runs generated with different seeds.

## 5.2 RESULTS

As shown in Figure 6, we find that neural synchrony, as measured by $SyncR^2$, exhibits a strong correlation with social performance within *Mistral* family, *Llama* family, and also cross-family *Mistral–Llama* pairs. Quantitatively, the correlation between $SyncR^2$ and the overall social performance is significant in all three family types (Mistral Family: Pearson $r = 0.88$, $p < 0.05$; Cross Family: Pearson $r = 0.89$, $p < 0.001$; Llama Family: Pearson $r = 0.99$, $p < 0.001$).

**Insights.** First, neural synchrony between LLMs serve as a robust indicator of social performance: model pairs with higher synchrony systematically achieve stronger social performance. Moreover, the relationship echoes neuroscience findings that inter-brain synchrony supports successful human social interaction, indicating a parallel between humans and LLMs in social interaction.

## 5.3 CONTROLLING FOR CONFOUNDING FACTORS

We note that more recent models cluster in the upper-right corner of the correlation plot, simultaneously exhibiting higher synchrony and better performance. This raises a potential concern: the observed relationship between neural synchrony and social performance might partly reflect confounding factors such as long-context processing capacity or instruction-following ability. In this case, better performance in SOTOPIA and higher neural synchrony could be attributed to the alignment on these general abilities rather than to genuine social capabilities. Our goal is therefore to test whether neural synchrony remains positively associated with social performance even after accounting for such confounds.

Table 1: Partial correlations between neural synchrony and social performance after controlling for different abilities. Rows correspond to model family types, and columns correspond to the datasets used for evaluation and control.

| Model Family | IFEval | MuSR |
|---|---|---|
| Mistral | 0.81 | 0.92 |
| Cross | 0.71 | 0.89 |
| Llama | 0.27 | 0.99 |

To address this, we incorporate two external evaluation datasets: IFEval (Zhou et al., 2023a), which assesses instruction-following ability, and MuSR (Sprague et al., 2023), which measures long-context reasoning. These abilities are crucial for success in SOTOPIA, which involves long interaction histories and explicit role-playing/formatting instructions. We then compute the partial correlations between neural synchrony and social performance while conditioning on these ability scores, to remove the correlations explained by these confounds. The results are shown in Table 1. After the control, the correlation between neural synchrony and social performance remains positive and significant, indicating that neural synchrony reflects social performance that cannot be attributed to other abilities, supporting its role as an indicator of the social dynamics between LLMs.

## 6 DISCUSSIONS

**Difference between representation similarity and neural synchrony in LLMs.** A key distinction is the requirement of sample correspondence. Previous approaches for measuring representation similarity require that two model backbones are exposed to the corresponding inputs: Canonical Correlation Analysis (CCA) (Hardoon et al., 2004; Raghu et al., 2017) searches for correlated projections under paired inputs, while Centered Kernel Alignment (CKA) (Kornblith et al., 2019) and Centered Kernel Nearest-Neighbor Alignment (CKNNA) (Huh et al., 2024) measure similarity by comparing the geometry of representations across corresponding samples. In this work, we explore neural synchrony between interacting LLMs, where each model receives different contextual information, shaped by its own background, personality, and social goals. These agent-specific inputs are inherently non-shared and lack direct correspondence, making the above similarity measures inapplicable. Our proposed neural synchrony evaluates how well the representations of one agent can anticipate the future representations of the other. It resolves the temporal dimension of the representations, allowing us to capture temporal dynamics that naturally arise during social interaction.

**Affine transformations as a minimal assumption for measuring neural synchrony.** Assuming that two socially interacting LLMs' representations can be connected through an affine transformation is a strong simplification. However, our results show that this simple assumption works well in practice. Across layer pairs of various interacting LLMs, the affine transformation consistently achieves high predictive performance and generalizes well to the test set (as examples shown in Figure 2). This suggests that affine transformations are able to capture meaningful information between the models' representations, and it is surprising that such a simple transformation is able to capture the social-related dynamics between interacting LLMs. The strong performance of affine transformations is also consistent with prior findings that LLM representations exhibit largely linear geometric structure across various domains. (Park et al., 2023; Li et al., 2023; Huh et al., 2024; Kim et al., 2025). To assess whether additional model capacity is necessary, we also evaluated a two-layer nonlinear transformation (detailed in Appendix I). These results indicate that greater expressive power does not consistently lead to better generalization of predictive performance. This supports the use of affine transformations as a simple, stable, and effective choice for measuring neural synchrony between socially interacting LLMs.

**Why does neural synchrony emerge? Links to Theory of Mind and social predictive coding.** There are several possible explanations for the emergence of neural synchrony between socially interacting LLMs. One explanation is that synchrony reflects implicit modeling of the partner's mental states. As an agent $A$ maintains internal representations of their own emotions, beliefs, desires, and intentions, its interacting partner $B$ may form parallel representations when reasoning about $A$'s mental states. This process resembles an implicit form of *Theory of Mind* (Birch & Bloom, 2004; Callaghan et al., 2005; Frith & Frith, 2005), where agents not only understand the dialogue but also reason about their partner's "mind" behind the dialogue. Another interpretation draws on *social predictive coding theory* (Tamir & Thornton, 2018; Thornton et al., 2019) in neuroscience, which posits that brains reduce uncertainty by predicting the future states in the social world; our affine transformations directly operationalize this idea at the latent representation level. These explanations are not mutually exclusive and may co-occur. If valid, they suggest that neural synchrony partly reflects these emergent cognitive or neural processes in LLMs during social interaction.

In additional experiments (detailed in Appendix J), we provide initial evidence that LLM representations encode **explicit** social states of others. We find that an agent's hidden representations encode *other's previous mental states*, i.e., emotions, while they are not directly observable to the

agent during interaction. This suggests an emergent form of Theory of Mind, where agents reason about other's mental states. Moreover, current representations contain information predictive of the partner's *future emotion* and *action* distributions, suggesting a form of social predictive coding where agents internally simulate how interactions may unfold. These findings further support our hypothesis that neural synchrony is connected to these key mechanisms of sociality in LLMs.

# 7 CONCLUSION

In this work, we introduced a novel framework for examining the "social minds" of LLMs by measuring neural synchrony between socially interacting agents. We defined $SyncR^2$ by training predictive affine transformations across extracted LLM representations. This provides a representation-level measure that captures the social temporal dynamics of LLM interaction, which is validated through controlled experiments. Our results have shown that this form of neural synchrony is strongly correlated with behavioral-level social performance, serving as an indicator for social dynamics between LLMs. These findings position neural synchrony as a novel proxy for studying the internal dynamics of LLM interactions, revealing surprising parallels with human inter-brain synchrony studies in neuroscience. We hope this work inspires future research toward a deeper understanding of the social aspects of LLMs, as well as how their social behaviors relate to internal representations. While our work uncovers correlations, suggests hypotheses about why they may arise, and explores initial findings of emergent social cognitive processes in LLMs, an important question is whether a causal relationship exists, which requires verification through intervention-based experiments. Moreover, our current experimental setting with SOTOPIA could be extended to longer-term, more open-ended, and even multi-agent scenarios, which would be more complex and realistic. In the future, we see opportunities to leverage neural synchrony to design more adaptive, cooperative, and socially intelligent LLM agents, where synchrony could serve as an indicator of agent intelligence during social interaction and guide training objectives.

ETHICS AND RISKS STATEMENT

Our study investigates what we term "neural synchrony" in language models during social interaction. We note that this terminology is metaphorical and refers solely to representational correlations between LLMs, rather than to biological neural activity. While this work does not directly involve human subjects or sensitive data, several ethical considerations remain. First, findings on existence of measured neural synchrony and its correlation with social performance may be misinterpreted as evidence of consciousness in LLMs. We emphasize that our results only demonstrate representational patterns in these LLM-based agents and do not imply that they develop human mental states. In addition, measuring "social minds" in LLMs with neural synchrony could be misused to anthropomorphize these systems, potentially leading to over-trust or inappropriate deployment in sensitive social or decision-making contexts. Such uses should be carefully scrutinized to avoid misleading users about the capabilities of these systems. We therefore call for cautious interpretation and responsible communication of these findings.

REPRODUCIBILITY STATEMENT

We have made every effort to ensure the reproducibility of our work. All implementation details, including model specifications, evaluation protocols, and hyperparameters for training the affine transformations, are described in the main text and appendix. The dataset (SOTOPIA) and open-sourced LLMs (Mistral and Llama families) we use are publicly available. Taken together, these resources ensure that our findings can be reliably reproduced and verified.

ACKNOWLEDGMENT

This work was partially supported by NSFC-6247070125. We thank the high-performance computing center at Eastern Institute of Technology and Institute of Digital Twin for providing the computing resources. We extend our gratitude to the reviewers for their insightful comments and valuable discussions.

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

## A    DETAILS AND EXAMPLES OF SOTOPIA AND SOTOPIA-EVAL

We present example SOTOPIA scenarios in Appendix A.1, example agent profiles in Appendix A.2, example interactions in Appendix A.3, and evaluation dimensions of SOTOPIA-EVAL in Appendix A.4.

### A.1    EXAMPLE SCENARIOS

**S1: "Music Decision"**
*Scenario.* Two friends are hanging out at home and deciding what music to listen to.
*Agent A's goal.* Listen to your favorite band (Extra information: your favorite band just released a new album).
*Agent B's goal.* Listen to a peaceful classical music to relax (Extra information: you had a stressful day and you just want to relax with some classical music).

**S2: "Loveseat Bargain"**
*Scenario.*Scenario: One person is offering an Italian Leather Loveseat for $50.0, while another person is showing interest to purchase it. The item is a full-sized loveseat, crafted from Italian leather in a faded burgundy color. It still has the original tags, and the current owner is the first and only owner. The transaction is strictly cash and carry as the owner does not offer delivery services.
*Agent A's goal.* You are the designated buyer for this item and your target price is set at $46.0. If you purchase this item at a significantly higher price than the target, you would face a penalty. However, if you manage to buy it for less than the target price, you will receive a bonus.
*Agent B's goal.* You are the seller for this item, with a target price of $26.5. You'll face a penalty if the item is sold significantly below this price. However, if you manage to sell it for more than the target price, a bonus will be your reward.

### A.2    EXAMPLE AGENT PROFILES

SOTOPIA character profiles include: name, pronouns, age, occupation; their Big-5 personality; moral and Schwartz values; decision style; public info; secret.

**P1: Mia (she/her)**

*Age/Job.* 29, Outdoor trip leader.
*Big-5.* High agreeableness, high conscientiousness, moderate neuroticism.
*Moral/Values.* Care, fairness; benevolence, universalism.
*Decision Style.* Behavioral.
*Public.* Always prepared; prefers clear boundaries.
*Secret.* Once had a panic attack sharing a tight tent space; avoids close sleeping proximity.

**P2: William (he/him)**

*Age/Job.* 30, Freelance photographer.
*Big-5.* High openness and extraversion.
*Moral/Values.* Loyalty, care; stimulation, hedonism.
*Decision Style.* Conceptual.
*Public.* Loves spontaneous trips.
*Secret.* Recently forgot to pack a sleeping bag; embarrassed to admit it.

### A.3    EXAMPLE INTERACTIONS

We provide example interactions in SOTOPIA between `Mistral-7B-Instruct-v0.3` and `Llama-3-8B-Instruct`.

**I1: Music Decision**

```
Turn 0. Amara Hartley said: "Hey Oliver, have you heard the new
album from my favorite band? I think it would be a great way to
unwind after a long day."
Turn 1. Oliver Thompson said: "I'm not really in the mood for
```

anything too loud or energetic today, how about some classical
music instead?"
Turn 2. Amara Hartley said:"That's cool, Oliver. Let's find
a peaceful classical piece that still captures the essence of
nature, like Vivaldi's Four Seasons."
Turn 3. Oliver Thompson said:"That sounds perfect, Amara!
I've been having a stressful day and some calming classical
music would really help me unwind. Let's go with Vivaldi's
Four Seasons, it's one of my favorites."
Turn 4. Amara Hartley said:"I'm glad we can agree, Oliver.
Let's enjoy the soothing melodies of Vivaldi's Four Seasons
while we catch up on our day."
...

**I2: Loveseat Bargain**

Turn 0. Micah Stevens said:"Hey Jaxon, I stumbled upon this
Italian leather loveseat on sale for $50.0. It's a full-sized
loveseat, ... What do you think about it?"
Turn 1. Jaxon Prentice said:"I'm intrigued by the description,
Micah. I've always been a fan of unique, one-of-a-kind pieces.
However, I'm a bit hesitant about the price. $50 seems a bit
steep for a used item, even if it's in good condition.
What makes you think it's worth that much?"
Turn 2. Micah Stevens said:"Well, Jaxon, considering it's crafted
from authentic Italian leather, has no tears or rips, and comes
with the original tags, I believe the price is fair. Plus, the
unique charm it adds to a Man Cave is priceless. But, I'm open to
negotiation if you're interested. What's your offer?"
Turn 3. Jaxon Prentice said:"I appreciate your willingness
to negotiate, Micah. I've got to be honest, $50 is still a bit
out of my budget. I was thinking more along the lines of $35 to
$40. The scratches from the cat do add a certain character to the
piece, but I'm not sure if it's worth the premium you're asking.
Can you meet me in the middle?"
Turn 4. Micah Stevens said:"Jaxon, I understand your concerns
about the price. However, the quality of Italian leather and the
unique character it has, due to the cat scratches, adds a level
of charm that's hard to find elsewhere. But, since we're friends,
I'm willing to meet you halfway. Let's say $46.0. It's a
fair price for a one-of-a-kind, authentic Italian leather
loveseat. What do you think?"
...
Turn 6. Micah Stevens said:"Jaxon, I respect your haggling skills,
but I think $46.0 is already a generous offer considering the
item's quality and rarity. However, since we're friends, I'll
sweeten the deal a bit. I'll bring the price down to $44.0. That's
a $9.0 discount from the original asking price. Plus, I'll throw
in the original tags and a small token of appreciation
as a friend. What do you say, my man? Let's seal this deal and
add this Italian leather loveseat to your Man Cave." (LEAVE)

## A.4   EVALUATION DIMENSIONS IN SOTOPIA-EVAL

There are a total of 7 evaluation dimensions in SOTOPIA-EVAL, listed as follows:

**Goal Completion [0–10]** is the extent to which the agent achieved their goals. Agents' social goals,
defined by the environment, are the primary drivers of their behavior.

**Believability [0–10]** focuses on the extent to which the agent's behavior is perceived as natural, realistic, and aligned with the agents' character profile, thus simulating believable proxies of human behavior.

**Knowledge [0–10]** captures the agent's ability to actively acquire new information. This dimension is motivated by the fact that curiosity, i.e., the desire to desire to know or learn, is a fundamental human trait.

**Secret [-10-0]** measures the need for agents (humans) to keep their secretive information or intention private.

**Relationship [-5–5]** captures the fundamental human need for social connection and belonging.

**Social Rules [-10–0]** concerns norms, regulations, institutional arrangements, and rituals.

**Financial and Material Benefits [-5–5]** pertains to traditional economic utilities as addressed by classic game theory.

## B    DATASET CONSTRUCTION FOR COMPUTING $SyncR^2(B \to A)$

To evaluate how well model $B$ predicts model $A$, we construct the dataset in a way analogous to Section 3.2, with the roles of $A$ and $B$ reversed. As in the main formulation, the task remains next-turn representation prediction. However, because $B$ acts after $A$ in each interaction round, the prediction target for $B$ is the representation of $A$ at the subsequent turn.

At each turn $t$, we collect the hidden representations $\boldsymbol{h}_{t,l_A}^{(A)}$ from agent $A$ at layer $l_A$ and $\boldsymbol{h}_{t,l_B}^{(B)}$ from agent $B$ at layer $l_B$. These temporally aligned pairs are used to construct the dataset:

$$\mathcal{D}_{l_B \to l_A}^{B \to A} = \left\{ \left( \boldsymbol{h}_{t,l_B}^{(B)}, \ \boldsymbol{h}_{t+1,l_A}^{(A)} \right) \ \Big| \ t = 1, \ldots, T-1 \right\}.$$

## C    SENSITIVITY ANALYSIS OF $SyncR^2$ TO LAYER CHOICE AND $R^2$ CLIPPING

### C.1    LAYER CHOICE IN COMPUTING $SyncR^2$

We use the best predictive match for each layer in model A when computing $SyncR^2$, as described in Section 3.2. This choice is motivated by findings in neuroscience showing that inter-brain synchrony arises between specific brain regions rather than uniformly across the entire brain (Dumas et al., 2010; Kawasaki et al., 2013). By analogy, we view these best predictive layer pairs as the most informative regions between two LLMs for measuring neural synchrony, whereas layer pairs with negative test-set $R^2$ are interpreted as exhibiting no synchrony.

To examine whether our conclusions depend on taking the average of only the best-matching layers, we compute $SyncR^2$ using the average of the top-$k$ predictive matches for each layer (with $k = 1$ corresponding to our original method). All other parts of the measurement remain unchanged. We then analyze how the correlation between $SyncR^2$ and social performance varies with different $k$s.

As shown in Figure 7, for the *Mistral* family, increasing $k$ strengthens the correlation between $SyncR^2$ and social performance. For the *Llama* and *Cross*-family model pairs, larger $k$ values slightly weaken this correlation. In all cases the correlation remains high ($r > 0.7, p < 0.05$). These results indicate that our original layer selection design is a reasonable choice, but not the only one: alternative selections lead to similarly strong correlations, showing that the metric and finding are robust across different design choices.

### C.2    CLIPPING OF NEGATIVE $R^2$ VALUES

To empirically assess the impact of setting negative $R^2$ values to zero, we compute $SyncR^2$ under two conditions: (1) using the original procedure where negative values are set to zero, and (2) keeping negative $R^2$ values. As shown in Figure 8, keeping negative $R^2$ values substantially reduces $SyncR^2$ and leads to markedly inflated standard errors of $SyncR^2$. This behavior is driven by the

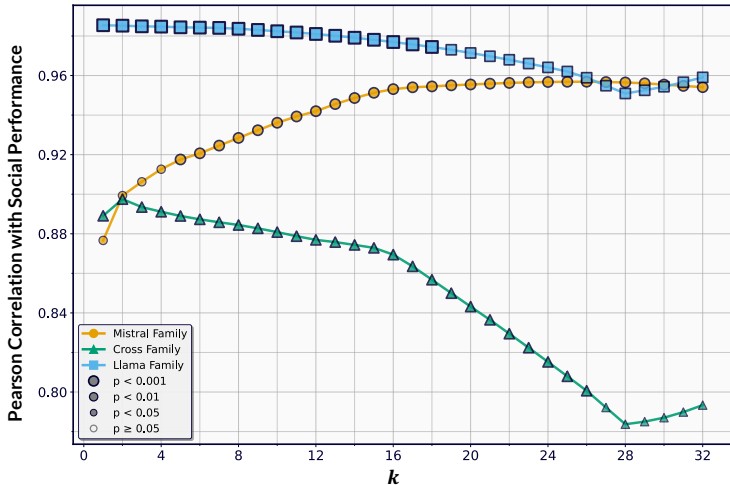

Figure 7: Pearson correlation between $SyncR^2$ (computed under the top-$k$ setting) and social performance as a function of $k$. The $x$-axis shows the number of top predictive matches ($k$, up to 32 layers). The $y$-axis shows the resulting Pearson correlation between $SyncR^2$ and social performance. Marker size indicates statistical significance, while shape and color denote the model family.

large variance introduced by negative $R^2$ values, which often arise when the predictive transformation fails to generalize. These results indicate that keeping negative $R^2$ leads to unstable and unreliable synchrony measurements.

## D    IMPACT OF SAMPLE SIZES ON NEURAL SYNCHRONY

We conducted a pilot experiment with different sample sizes (Figure 9). The results show little improvement beyond 6500 samples. Moreover, setting the size to 7500 would leave some model pairs without enough data within three simulations in SOTOPIA, not ensuring a fair comparison. We therefore fix the sample size at 6500 for all models to ensure a fair comparison.

## E    DETAILS OF PERSONA SAMPLING AND ITS EFFECT ON NEURAL SYNCHRONY

In SOTOPIA, persona pairs are not fixed across scenarios. There are a total of 90 scenarios, and each scenario is paired with 5 persona pairs chosen from a pool of 40 different personas. This results in 450 interaction tasks (90 scenarios × 5 persona pairs) where LLMs take on different persona pairs in different scenarios. By ensuring "representations from the same interaction are not split between train and test sets", we guarantee that a representation pair in the test set never shares the same scenario–persona pair as any representation pair in the train set, meaning that affine transformation cannot rely on memorizing specific scenarios–persona pairs when evaluated on the test set.

While there are no identical scenario-persona pairs, there are indeed identical agent pairs paired to different scenarios in the Sotopia data. To rule out the effect of shared persona pairs between sets, we conduct experiments with a new sampling implementation, ensuring that the same persona pairs do not appear in both the test and train sets. All other settings remained unchanged, and we trained and evaluated affine transformations for three representative model pairs from each family type. As shown in Table 2, the $SyncR^2$ values under the new sampling remain largely unchanged compared to the original sampling, suggesting that it is not the shared persona pairs that drive neural synchrony.

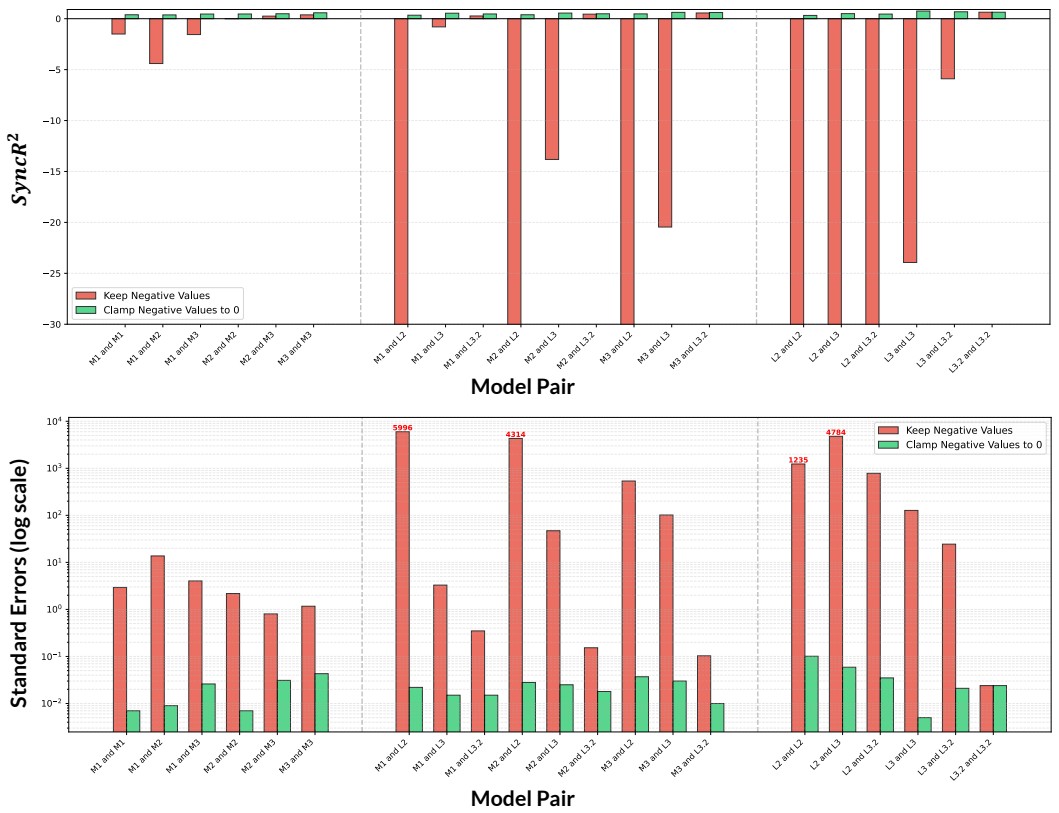

Figure 8: Comparison of $SyncR^2$ computed with and without clamping negative values. Labels on the *x*-axis correspond to model pairs (model names are abbreviated; see Appendix F for the full name correspondence), grouped by their model family. The upper panel reports $SyncR^2$ values; the *y*-axis is truncated at –30 to exclude extreme outliers for visibility. The lower panel reports the standard errors of $SyncR^2$ on a logarithmic scale.

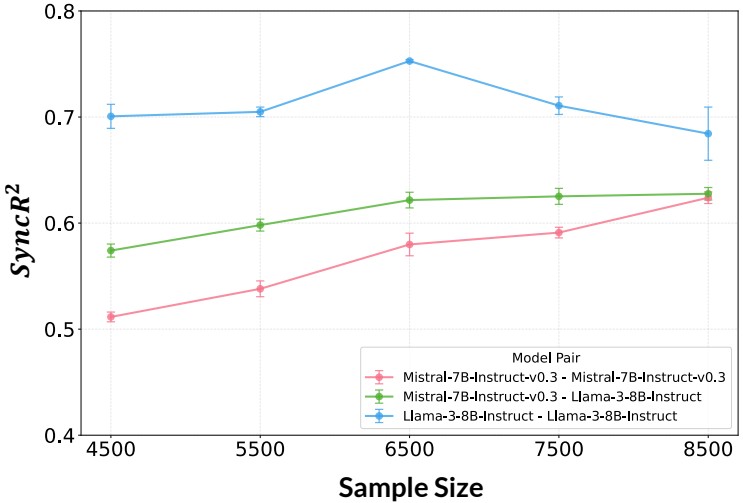

Figure 9: $SyncR^2$ as a function of dataset sample size for training affine transformations. Error bars indicate standard errors.

Table 2: Comparison of $SyncR^2$ under original and new sampling implementations for the train and test split. In the new sampling, we guarantee that no same persona pair appears in both sets.

| Model Pair | Original | New |
|---|---|---|
| `Mistral-7B-Instruct-v0.2` & `Mistral-7B-Instruct-v0.2` | $0.46 \pm 0.01$ | $0.47 \pm 0.02$ |
| `Mistral-7B-Instruct-v0.2` & `Llama-3-8B-Instruct` | $0.55 \pm 0.02$ | $0.58 \pm 0.02$ |
| `Llama-3-8B-Instruct` & `Llama-3-8B-Instruct` | $0.75 \pm 0.01$ | $0.74 \pm 0.05$ |

Table 3: Correspondence between abbreviations and full model names.

| Abbreviation | Full Name |
|---|---|
| M1 | Mistral-7B-Instruct-v0.1 |
| M2 | Mistral-7B-Instruct-v0.2 |
| M3 | Mistral-7B-Instruct-v0.3 |
| L2 | Llama-2-7B-Instruct |
| L3 | Llama-3-8B-Instruct |
| L3.2 | Llama-3.2-3B-Instruct |

# F CORRESPONDENCE BETWEEN ABBREVIATIONS AND FULL MODEL NAMES

We present the mappings of abbreviation to full names in Table 3.

# G IMPLEMENTATION DETAILS FOR INVESTIGATING HOW DIFFERENT RELATIONSHIPS SHAPE SYNCHRONY

We partition the test set according to the predefined relationship categories in SOTOPIA: *Strangers*, *Know-by-names*, *Acquaintances*, *Friends*, *Romantic Relationships*. Using the affine mappings learned during training, we then re-evaluate the model pairs on these relationship-specific subsets of the test set. The synchrony score $SyncR^2$ is then computed following the same procedure as in the main experiments, for each subset of relationships. When comparing different relationships, we subsample equal-sized subsets and report the average over three random seeds to reduce potential biases.

Notably, to ensure a fair comparison, we focus only on scenarios drawn from *social iqa*, *social chemistry*, and *normbank*. These scenarios focus on everyday social encounters and contain multiple and even relationship types within the same scenario. In contrast, scenarios such as *mutual friends* (cooperative) or *craigslist bargain* (competitive) implicitly fix the relationship as "strangers," which would confound our analysis.

# H COMPARISON OF GPT-OSS-120B EVALUATIONS AND HUMAN EVALUATIONS

We compare the evaluations of `gpt-oss-120b` with human evaluations using the data collected in (Zhou et al., 2023c). The quantitative results are summarized in Table 4 and Figure 10.

We adopt `gpt-oss-120b` as the LLM evaluator for three main reasons. First, as shown in Table 4, the Pearson correlations between `gpt-oss-120b` and human judgments are on par with those of GPT-4, and even higher on several dimensions (e.g., *Secret* and *Believability*). Second, Figure 10 shows that 75.8% of the scores produced by `gpt-oss-120b` fall within one standard deviation of human ratings, indicating stable agreement. Finally, compared to expensive models such as GPT-4, `gpt-oss-120b` is significantly more cost-efficient. This efficiency allows us to avoid sampling and instead evaluate on the full set of SOTOPIA simulations for all model pairs, leading to more accurate and convincing results. Overall, these comparisons support the use of `gpt-oss-120b` as a reliable and practical evaluator in our experiments.

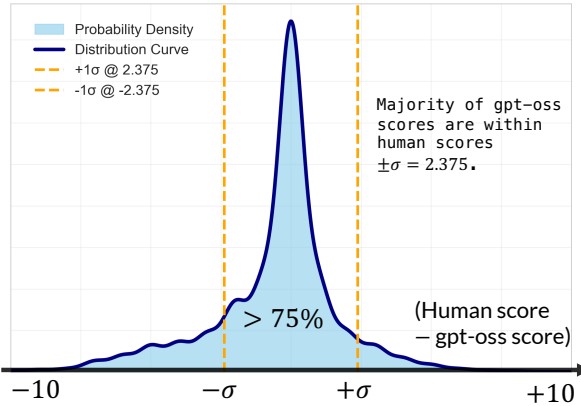

Figure 10: Distribution of the difference between the scores given by humans and `gpt-oss-120b`.

Table 4: Pearson correlation coefficients and $p$-values between different LLM evaluators' evaluation and human judgment on **models'** output among different dimensions. Strong and significant correlations are in blue.

| Dim. | GPT-4 | gpt-oss-120b |
|---|---|---|
| Secret | 0.22** | 0.46** |
| Knowledge | 0.33** | 0.28** |
| Social Rules | 0.33** | 0.22* |
| Believability | 0.45** | 0.50** |
| Relationship | 0.56** | 0.45** |
| Financial Benefits | 0.62** | 0.38** |
| Goal Completion | 0.71** | 0.60** |
| ** : $p \leq 0.01$, * : $p \leq 0.05$ | | |

## I  RESULTS OF NONLINEAR TRANSFORMATIONS

To explore nonlinear transformations for measuring neural synchrony, we compare the performance of nonlinear transformation with the affine transformation across three model pairs from each family type. We adopt a two-layer nonlinear transformation with a hidden dimension of 512 and ReLU activation. For training, we use AdamW with a learning rate of 1e-4, a weight decay of 0.1, and train for 200 epochs.

**Results**. As shown in Table 5, nonlinear transformations offer limited improvement for (`Mistral-7B-v0.2`, `Mistral-7B-v0.2`) and (`Mistral-7B-v0.2`, `Llama-3-8B-Instruct`) pairs, and perform notably worse than the affine transformation for (`Llama-3-8B-Instruct`, `Llama-3-8B-Instruct`). This suggests that greater expressive power does not consistently lead to better generalization of predictive performance in our experiments.

## J  EMERGENT THEORY OF MIND AND SOCIAL PREDICTIVE CODING OF LLMS IN SOCIAL INTERACTION

**Setting and implementation details.** We run additional simulations in SOTOPIA, where each agent is instructed to explicitly output its current emotion and action type at the beginning of every turn. Emotion categories are based on Plutchik's Wheel of Emotions (Plutchik, 1982), covering eight primary emotions: *Joy, Trust, Fear, Surprise, Sadness, Disgust, Anger, Anticipation*. Action types are based on Searle's taxonomy of speech acts (Searle, 1975; 1979), including *Assertive, Directive, Commissive, Expressive, and Declaration*. Note that the partner's interaction history excludes the explicit emotion and action outputs, so that they are never directly observed by the other agent. For each turn, we extract the token-level logits corresponding to the emotion and action outputs,

Table 5: Comparison of nonlinear and affine transformations in measuring neural synchrony.

| Model Pair | Affine | Nonlinear |
|---|---|---|
| `Mistral-7B-Instruct-v0.2` & `Mistral-7B-Instruct-v0.2` | $0.46 \pm 0.01$ | $0.47 \pm 0.01$ |
| `Mistral-7B-Instruct-v0.2` & `Llama-3-8B-Instruct` | $0.55 \pm 0.02$ | $0.60 \pm 0.01$ |
| `Llama-3-8B-Instruct` & `Llama-3-8B-Instruct` | $0.75 \pm 0.01$ | $0.59 \pm 0.01$ |

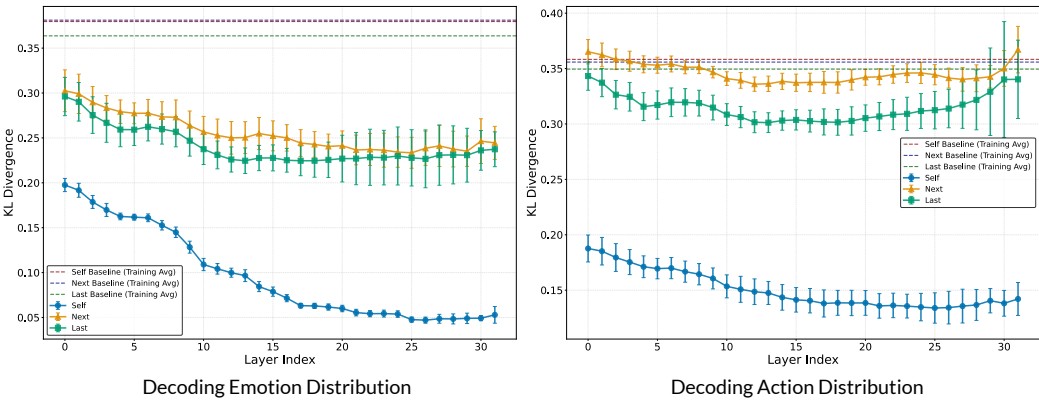

Figure 11: Decoding action and emotion distributions from current, previous, and next agent representations. The y-axis indicates test-set KL divergence (lower is better). Baselines are computed using the average emotion/action distributions from the training set. Results are averaged over three random seeds, with error bars showing standard errors.

and use them as target distributions. We then test whether these distributions can be decoded from the hidden representations of itself and the other agent. We conduct these experiments with `Mistral-7B-Instruct-v0.3` and `Llama-3-8B-Instruct`, as these models reliably follow the instruction format. Other models are less capable and often produced broken outputs, therefore were excluded from this experiment.

> **Additional Instruction**
>
> Please be sure to indicate your emotion and action type both in a word at the beginning of your turn, in the format of: Emotion. Action type. Response. Emotion should be one of the following words: Joy. Trust. Fear. Surprise. Sadness. Disgust. Anger. Anticipation.; Action type should be one of the following words: Assertive. Directive. Commissive. Expressive. Declaration. Your output format should be: Emotion. Action type. Response.

**Results.** Figure 11 summarizes the decoding performance. For emotion distributions, models achieve high performance when decoding their own current emotions, which serves as a sanity check. Crucially, decoding of both the partner's previous emotions and the partner's future emotions is also significantly above chance, suggesting that agents' representations carry information predictive of others' mental states. For action distributions, decoding performance is again high for self actions, but the ability to decode future actions is only slightly above random baselines. These findings indicate that LLMs in social interaction may emerge Theory of Mind, where they actively understand their partners' mental states. They may also predict future emotions and actions which posits possibility for a social predictive coding, where they actively predict future social world states.

## K  THE USE OF LARGE LANGUAGE MODELS (LLMs)

In preparing this manuscript, we made limited use of the LLMs. The LLMs were used solely to aid in revising and polishing the writing for clarity and readability. All ideas, experimental design, data collection, analysis, and interpretation were conducted entirely by the authors. The authors

reviewed and verified all text polished by the LLMs to ensure accuracy and alignment with the intended meaning.

