# OpenReview forum: "Neural Synchrony Between Socially Interacting Language Models"
_ICLR.cc/2026/Conference — ICLR 2026 Poster_

### Official Review · Reviewer_xLjv · 2025-11-01

**Soundness:** 3
**Presentation:** 3
**Contribution:** 3
**Rating:** 6
**Confidence:** 3

**Summary:**

This paper introduces a novel framework for measuring neural synchrony between socially interacting large language models (LLMs), drawing inspiration from neuroscience research on inter-brain synchrony in humans. The authors simulate multi-turn social interactions using SOTOPIA and extract hidden representations from paired LLM agents, then train predictive affine transformations to measure how well one agent's current representations can predict another's future representations, quantified by a metric called SyncR². Through rigorous control experiments across 21 model pairs from the Mistral and Llama families, they demonstrate that measured synchrony genuinely reflects both social engagement and temporal alignment, and remarkably, this representation-level neural synchrony strongly correlates with behavioral-level social performance (Pearson r = 0.88-0.99, p < 0.05) even after controlling for confounding factors like instruction-following and long-context reasoning abilities. These findings reveal striking parallels between LLMs and human brains in social interaction dynamics, suggesting that LLMs may possess internal mechanisms analogous to human inter-brain synchrony that support successful social coordination, thereby offering the first empirical evidence for examining the "social minds" of LLMs at the representational level.

**Strengths:**

1. This paper is the first to apply the neuroscience concept of "inter-brain synchronization" (IBS) to LLMs, which is highly innovative.
2. The experimental setup was very thorough, and the comparison results are quite convincing. The authors constructed 21 model pairs (covering two major families) and simulated 450 scenarios × 3 random seeds.

**Weaknesses:**

1. As the author mentioned, there is a lack of validation for larger models, such as the 14b and 32b models.
2. SOTOPIA has certain limitations, including only short-term interactions (averaging 6-8 rounds) and institutionally structured communication.
3. It is possible to explore more agents than just two, which would better simulate real-world communication scenarios.
4. The control group setup in the article still has some shortcomings; more control groups could be added to isolate scene and agent factors and verify the necessity of reciprocity, among other things.

**Questions:**

1. Regarding the results shown in Figures 2 and 3. Why are the areas indicated by the black boxes not clustered around the diagonal line?
2. Why use "affine transformation" (linear mapping + bias)? The representation space of LLMs may be highly nonlinear.
3. What is the meaning of a negative R² value?

---

> ### Author Response · Authors · 2025-11-21
> **Response to Reviewer xLjv (Part 1/3)**
>
> We thank reviewer xLjv for the insightful feedback. We address the questions and concerns below.
>
> > **W1. Lack of larger models.**
>
> Thank you for raising this concern. While these models are not included in our current experiments, our study already includes 21 model pairs with a diverse range of models from the *Mistral*, *Llama*, and *Cross-model* families, with multiple versions and parameter sizes. The exclusion of larger models is primarily due to their high computational demands.
>
> We acknowledge that larger models could introduce new complexities, and validating our findings across these models will be a key next step. We view our results as an important initial step, presenting meaningful findings from a neuroscience perspective, and providing a foundation for future work that can expand to larger models and broader model families.
>
> > **W2 & W3. Limitations of Sotopia; exploring multiple agents.**
>
> We appreciate the reviewer’s observation regarding the limitations of Sotopia, including its short interaction length, structured communication, and two-agent setting. While these constraints are inherent for the environment, Sotopia still provides a diverse range of social scenarios, detailed and realistic agent profiles, and fine-grained, interpretable evaluation metrics, making it *well-suited for a controlled analysis of LLM social capabilities*. The structured communication format also enhances experimental control, ensuring more reliable results. We agree that extending to longer-term, less constrained, and multi-agent environments is a valuable direction, and we have included this as an important direction for future work in the revised manuscript.
>
>
>
> > **W4. More control groups to isolate scene and agent factors.**
>
> Thank you for the thoughtful comment. We agree that establishing appropriate control conditions is important. In our study, Control Group 1 investigates the role of social engagement, while Control Group 2 investigates the role of temporal proximity. Since Control Group 2 still uses representations from within the same interaction, it also rules out potential confounds such as static representational similarity and shared context.
>
> We additionally conducted experiments to exclude interference from sharing the same scenario, i.e, negotiation. We used all 50 negotiation tasks in Sotopia and sampled 15 interactions with different seeds for each task. Due to time constraints, we used Mistral-7B-Instruct-v0.3 for both agents for its fast generations.
>
> We computed neural synchrony for the experimental condition using the same method as in the paper. We then introduced a new control, Control Group 3. In this control, we split the interactions into two halves and paired agent interactions across halves, producing pairs such as (A, B) and (C, D) where A and C share the same LLM, as do B and D. They share the same type of scenario which is negotiation. We then trained affine transformations for representations from A to predict next representations from D, and representations from C to predict next representations from B. Since our synchrony measure is bidirectional, we also trained transformations from D and B predicting corresponding "next" representations, and averaged the results.
>
> The standard representation pairing method used in the experimental group yields a $SyncR^2$ of 0.53 with a standard error of 0.02. In contrast, Control Group 3 yields a $SyncR^2$ of 0, with all transformations failing to generalize to the test set better than the mean. This indicates that models cannot predict the next representations of agents who are not in the same interaction solely based on shared negotiation incentives. The synchrony we observe therefore reflects adaptation within the specific interaction, rather than being driven by the shared scenario context.
>
> Apart from these three control groups, if there are any other factors you believe should be further controlled for, we would greatly appreciate your suggestions and would be happy to conduct additional control experiments to address them.

---

> ### Author Response · Authors · 2025-11-21
> **Response to Reviewer xLjv (Part 2/3)**
>
> > **Q1. Why are the areas indicated by the black boxes not clustered around the diagonal line in Figures 2 and 3?**
>
> Thank you for the insightful question. The fact that the best-predicting layer pairs (black boxes) are not clustered along the diagonal is expected, and informative. If neural synchrony were primarily capturing structural representational similarity, we would expect same-family pairs (e.g., Mistral-7B-Instruct-v0.2 predicting Mistral-7B-Instruct-v0.2 in Figure 2) to show strongest matches along the diagonal, but they do not.
>
> For this particular Mistral-7B-Instruct-v0.2 example, we observe that layers around 10–15 are most predictable from many source layers. Prior work on Theory-of-Mind–related representations, specifically in Mistral-7B, has reported that the same layer range are where the model encodes others’ beliefs most strongly during social reasoning [1]. This aligns with our interpretation that neural synchrony captures potential Theory of Mind processes within LLMs.
>
> > **Q2. Why use affine transformation while the representation space of LLMs could be highly nonlinear?**
>
> Thank you for your thoughtful feedback. We agree that assuming a linear relationship between model representations is a simplification. However, our results show that this simple assumption works well in practice. Across layer pairs of various interacting LLMs, the affine transformation consistently achieves high predictive performance and generalizes well to the test set (as examples shown in Figure 2).
> This suggests that affine transformations are able to capture meaningful information between the models’ representations, and it is surprising that such a simple transformation is able to capture the representational alignment between interacting LLMs.
> The strong performance of affine transformations aligns with prior works suggesting that LLM representations exhibit linear geometric structure across various tasks [2,3,4].
>
> To further examine whether additional model capacity is beneficial, we evaluated a two-layer MLP with a hidden dimension of 512 and ReLU activation. For training, we use AdamW with a learning rate of 1e-4, a weight decay of 0.1, and train for 200 epochs. Across three representative model pairs from each family type, we find that the nonlinear transformation yields only marginal improvements or worse generalization performance, as shown in Rebuttal Table 1.
>
> These results suggest that increasing expressiveness does not necessarily improve generalization. This supports the use of affine transformations as a simple, stable, and effective choice for measuring neural synchrony between socially interacting LLMs.
> We have added the new discussion and results in Section 6 and Appendix I of the revised manuscript.
>
> > **Q3. What is the meaning of a negative $R^2$ value?**
>
> A negative $R^2$ value indicates that the learned transformation fails to generalize on the test set better than the mean. Formally, for each sample $i$ and output dimension $j$,
> $
> R\_j^2 = 1 - \\frac{\\sum\_i{(y\_{ij} - \\hat{y}\_{ij})^2}}{\\sum\_i{(y\_{ij}- \\bar{y}\_j)^2}}
> $, and
> $
> R^2=\\frac{1}{d}\\sum^d\_{j=1}{R^2\_j}
> $.
> A constant mean predictor $\hat{y_{ij}}=\bar{y_j}$ achieves $R^2_j=0$. $R^2<0$ indicates that learned transformation performs worse than predicting the mean.
>
>
> **Rebuttal Table 1**: Comparison of nonlinear and affine transformations in measuring neural synchrony.
> | **Model Pair**                                   | **Affine**        | **Nonlinear**    |
> |--------------------------------------------------|-------------------|------------------|
> | Mistral-7B-Instruct-v0.2 & Mistral-7B-Instruct-v0.2 | 0.46 ± 0.01       | 0.47 ± 0.01      |
> | Mistral-7B-Instruct-v0.2 & Llama-3-8B-Instruct       | 0.55 ± 0.02       | 0.60 ± 0.01      |
> | Llama-3-8B-Instruct & Llama-3-8B-Instruct            | 0.75 ± 0.01       | 0.59 ± 0.01      |

---

> ### Author Response · Authors · 2025-11-21
> **Response to Reviewer xLjv (Part 3/3)**
>
> **References**
>
> [1] Zhu, Wentao, et al. "Language Models Represent Beliefs of Self and Others." International Conference on Machine Learning. PMLR, 2024.
>
> [2] Park, Kiho, et al. "The Linear Representation Hypothesis and the Geometry of Large Language Models." International Conference on Machine Learning. PMLR, 2024.
>
> [3] Li, Kenneth, et al. "Inference-time intervention: Eliciting truthful answers from a language model." Advances in Neural Information Processing Systems 36 (2023): 41451-41530.
>
> [4] Kim, Junsol, et al. "Linear Representations of Political Perspective Emerge in Large Language Models." The Thirteenth International Conference on Learning Representations.

---

### Official Review · Reviewer_m5ti · 2025-11-01

**Soundness:** 3
**Presentation:** 3
**Contribution:** 2
**Rating:** 6
**Confidence:** 4

**Summary:**

This paper proposes to study the neural synchrony between language models which are engaged in a social interaction, inspired by the phenomenon of inter-brain synchrony in humans. They measure neural synchrony between interacting LMs by proposing a metric Sync$R^2$, which is based on predicting the other agent’s next response based on an affine transformation of the agent’s current representation of the conversation. They validate this metric and find that it reflects both social engagement and temporal alignment in their interactions and is strongly correlated with their social performance.

**Strengths:**

**S1**: This paper introduces an interesting and to me novel perspective on studying LM social capabilities through the lens of neural synchrony, rather than through behavioural outputs alone. I think that the motivation and connection with neuroscience is interesting and brings a fresh perspective. Although it may seem anthropomorphic on first glance, I do think it also makes sense to study this property in interacting LMs and the proposed metric could provide another tool for probing LMs.

**S2**: The validation approach is thorough and tries to account for many potential confounders, such as common context. I think the controls demonstrate to a satisfactory extent that synchrony is not due to other factors. Using models from different families and of different sizes also strengthens the conclusions.

**S3**: The paper is generally well-structured and clear with a good motivation, clear figures and a good progression from section to section. The connection to neuroscience is also explained clearly for an ML audience.

**Weaknesses:**

**W1**: Although I like the current ablations, I think there are still some potential issues that I can see with the setup. I think due to the strong claim (that LMs do exhibit neural synchrony), I would want to see some strong evidence ruling out alternative explanations.

- **W1a**: One is that the paper states that "representations from the same interaction are not split between train and test sets", but it doesn't seem clear whether representations from the same personas in different scenarios could appear in both sets, or if the persona pairs are randomized (apologies if I missed this). Because agent personas have consistent character backgrounds, it seems possible that the affine transformation could be learning identity mappings between personas rather than displaying true neural synchrony.

- **W1b**: I like the passive control, but I think further experiments could help disentangle genuine synchrony from shared context? What I mean is that both of the interacting models could be encoding some information such as ("this is a negotiation", "the negotiation is over buying a TV", "price range 300-400", "need to reach agreement") etc. This means that apparent synchrony could just be because both agents are reading the same scenario and encoding the constraints of that scenario. I think to see if this is due to a shared scenario like the negotiation I talked about vs. a true adaptation to the other agent, it would be good to test whether Sync$R^2$ drops when agents are in different interactions that are similar in type (e.g. the TV negotiation) but they don't actually interact with each other. E.g. A <-> B, and C <-> D could be two agent pairs in the same scenario, and we want to see the predictive power of A on D, B on C etc. If the metric is lower in this case I think this would be a stronger indication of synchronization within that interaction.


**W2**: The mechanistic explanation for this remains limited, as while the paper demonstrates that synchrony occurs and correlates with performance, the explanation of why this should be the case is somewhat superficial. There is some discussion of theory and mind and predictive coding, and Appendix F is interesting, but I think it would strengthen the paper significantly to expand on this section and add it to the main paper. For instance, are layers with higher Sync$R^2$ also higher in decoding performance, and which way does the causal relationship go?

**Questions:**

- Can you elaborate more on how the agents and interactions were sampled and split between the train/test sets?

---

> ### Author Response · Authors · 2025-11-21
> **Response to Reviewer m5ti (Part 1/3)**
>
> We thank reviewer m5ti for the insightful feedback. We address the questions and concerns below.
>
> > **W1(a) & Q1. Same personas could appear in both train and test sets; details of data split sampling; the affine transformations could be learning identity mappings between personas.**
>
> We appreciate the reviewer’s careful consideration of this point. In Sotopia, persona pairs are not fixed across scenarios. There are a total of 90 scenarios, and each scenario is paired with 5 persona pairs chosen from a pool of 40 different personas. This results in 450 interaction tasks (90 scenarios $\times$ 5 persona pairs) where LLMs take on different persona pairs in different scenarios.
>
> By ensuring "representations from the same interaction are not split between train and test sets", we guarantee that a representation pair in the test set never shares the same scenario–persona pair as any representation pair in the train set. This ensures that the affine transformation cannot rely on memorizing specific scenarios–persona pairs when evaluated on the test set.
>
> While there are no identical scenario-persona pairs, there are indeed identical agent pairs paired to different scenarios in the Sotopia data, as the reviewer pointed out.
> However, if the transformation were only learning identity mappings between personas, the temporal-lag control (control 2) should remain stable across lag values, since *persona information is identical for representations in the same interaction*. Instead, we observe a collapse as lag increases, indicating that the neural synchrony arises with turn-level temporal proximity instead of the same personas within an interaction.
>
> To further verify this, we conducted additional experiments with a new sampling implementation, ensuring that the same persona pairs do not appear in both the test and train sets. All other settings remained unchanged, and we trained and evaluated affine transformations for three representative model pairs from each family type. As shown in Rebuttal Table 2, the $SyncR^2$ values under the new sampling remain largely unchanged compared to the original sampling, suggesting that it is not the shared persona pairs that drive neural synchrony.
>
> We appreciate the opportunity to clarify this and have included the clarification and additional experiments in the Appendix E of the revised manuscript.
>
> **Rebuttal Table 2**: $SyncR^2$ under different sampling implementations for the train and test split.
> | Model Pair                                           | Original Sampling | New Sampling (No same persona pairs in both sets) |
> |------------------------------------------------------|-------------------|----------------------------------------------------|
> | Mistral-7B-Instruct-v0.2 and Mistral-7B-Instruct-v0.2 | 0.46 ± 0.01       | 0.47 ± 0.02                                        |
> | Mistral-7B-Instruct-v0.2 and Llama-3-8B-Instruct     | 0.55 ± 0.02       | 0.58 ± 0.02                                        |
> | Llama-3-8B-Instruct and Llama-3-8B-Instruct         | 0.75 ± 0.01       | 0.74 ± 0.05                                        |

---

> ### Author Response · Authors · 2025-11-21
> **Response to Reviewer m5ti (Part 2/3)**
>
> > **W1(b). Genuine synchrony should be distinguished from shared context.**
>
> Thank you for the thoughtful suggestion. We agree that shared context is a plausible alternative explanation for neural synchrony. However, we believe that our Control Group 2 addresses this concern. In this control, the interaction and scenario remain the same, but we pair representations with longer time lags instead of the most proximate turns. Since representations in the same interaction encode the same shared context, if synchrony were solely driven by shared context, neural synchrony should remain stable across different lags. Instead, we observe a collapse in synchrony as time lag increases. This indicates that neural synchrony is not merely driven by shared context.
>
> Following your suggestion, we also conducted an additional validation experiment. We used all 50 negotiation tasks in Sotopia and sampled 15 interactions with different seeds for each task. Due to time constraints, we used Mistral-7B-Instruct-v0.3 for both agents for its fast generations.
>
> We computed neural synchrony for the experimental condition using the same method as in the paper. We then introduced a new control, Control Group 3. In this control, we split the interactions into two halves and paired agent interactions across halves, producing pairs such as (A, B) and (C, D) where A and C share the same LLM, as do B and D. They share the same type of scenario which is negotiation. We then trained affine transformations for representations from A to predict next representations from D, and representations from C to predict next representations from B. Since our synchrony measure is bidirectional, we also trained transformations from D and B predicting corresponding "next" representations, and averaged the results.
>
> The standard representation pairing method used in the experimental group yields a $SyncR^2$ of 0.53 with a standard error of 0.02. In contrast, Control Group 3 yields a $SyncR^2$ of 0, with all transformations failing to generalize to the test set better than the mean. This indicates that models cannot predict the next representations of agents who are not in the same interaction solely based on shared negotiation incentives. The synchrony we observe therefore reflects adaptation within the specific interaction, rather than being driven by the shared scenario context.

---

> ### Author Response · Authors · 2025-11-21
> **Response to Reviewer m5ti (Part 3/3)**
>
> > **W2. Limited mechanistic explanations; expanding Appendix F.**
>
> Thank you for the insightful question. We agree that identifying a causal relationship between neural synchrony and the emergent Theory of Mind and/or Social predictive coding would be a very significant contribution.
>
> Appendix F (Appendix J in the revised manuscript) presents initial evidence of emergent Theory of Mind and Social predictive coding in LLMs during social interaction. However, a main limitation is that, with the current experimental setup, we are unable to quantify or establish a causal relationship between neural synchrony and the emergent social cognitive processes. The experiments in Appendix F (Appendix J in the revised manuscript) involve complex instruction following due to the additional prompt, and only two models currently generate reliable outputs under this setup. A more comprehensive quantitative analysis would require additional experiments, such as causal interventions, as well as a broader set of models capable of robustly following these instructions, which is unfortunately not feasible for us at this stage. We view this as a very important and promising direction for future research and have highlighted it in the future work section in the revised manuscript.
>
> That said, we would still like to offer additional interpretations of the Appendix F (Appendix J in the revised manuscript) results, inspired by your insightful question, which may motivate future research directions:
>
> - In the left panel of Figure 11 (in the revised manuscript), we observe that the model (1) forms a stable understanding of the other’s *past* emotions around Layer 12. Here, “stable” means that the KL divergence (as a measure of decoding performance) stops decreasing notably beyond this point. The model then (2) begins to stably predict other’s *future* emotions around Layer 21, and (3) forms a stable prediction of its *own* emotion around Layer 25. This pattern may suggest an internal sequence (*past* $\to$ *future* $\to$ *present*) in which the model first infers other's mental states based on their actions (Theory of Mind), then uses this understanding to predict others' future mental states (social predictive coding), and finally builds on these predictions to plan its own actions. This interpretation proposes a possible order of these emergent internal cognitive processes during social interactions.
>
> - When comparing the right panel of Figure 11 with the left panel of Figure 3 which share the same pair of models, we observe a relationship between neural synchrony and decoding performance. In Figure 3, after Layer 20, Mistral-7B-Instruct-v0.3 largely fails to predict the future representations of Llama-3-8B-Instruct. In Figure 9, we observe a *similar decline in performance* when predicting others' actions after Layer 20, as indicated by the increasing KL divergence.
> Moreover, the decoding performance of its *own* actions becomes progressively better from Layer 0 to Layer 20 and then stabilizes. Together, these patterns may suggest that around Layer 20, the model may shift from refining predictions about the other agent to focusing on formulating its own response, which aligns with the observed drop in both neural synchrony and the decoding performance of others.

---

### Official Review · Reviewer_tXna · 2025-11-01

**Soundness:** 3
**Presentation:** 4
**Contribution:** 3
**Rating:** 4
**Confidence:** 4

**Summary:**

This paper investigates representational dynamics between interacting LLMs in social simulation environments. The authors train affine transformations to predict one agent's future hidden representations (timestep t+1) from another agent's current representations (timestep t), defining a metric called $SyncR^2$ based on prediction accuracy. Experiments across 21 model pairs show that $SyncR^2$ decreases without active engagement or temporal alignment, and correlates positively with social task performance. Additional analyses reveal higher $SyncR^2$ in closer agent "relationships" and evidence that representations encode partners' mental states.

**Strengths:**

S1. This work is novel in exploring whether LLMs exhibit similarity at the representation level during social interaction, and designs a new evaluation method (whether these representations can be used to predict the next behavior of the interacting agent) to examine the existence of such social minds.

S2. I must say that the paper is very clearly written. This work allows readers unfamiliar with inter-brain synchrony background to easily understand the intention and existing work. The paper provides detailed and reasonable discussion on how to avoid overclaiming through rigorous experimental design (such as discussion of potential confounding factors) and ablation studies.

S3. The paper obtains some practically meaningful findings, such as discovering that neural synchrony is significantly correlated with social performance, and that relationship closeness affects synchrony strength. This is meaningful for future research on agent "social minds".

**Weaknesses:**

W1. The choice of affine transformation lacks good justification (why should the relationship between representations be linear?). Although the results show this transformation is meaningful for using one LLM's representation at time t to predict another LLM's representation at time t+1, there is no theoretical explanation, nor are alternatives provided.

Additionally, the explanation for why predictability can measure synchrony is incomplete. The current logic of the paper is: IBS is analogized to LLM representational predictability, predictability can measure synchrony, and this ability can explain social interaction performance. However, predictability does not equal synchrony. Synchrony in neuroscience is bidirectional, while LLM synchrony is unidirectional prediction from LLM A(t) to LLM B(t+1). An extreme case is: suppose a strong model A can predict weak model B's behavior, but B cannot predict A, then the final measure SyncR^2 cannot reach a high level, and synchrony does not exist here. On the other hand, regarding the time window, the choice of t->t+1 is not necessarily reasonable. According to strict synchrony definition, would examining t->t correspondence be more appropriate? Control 2 examined t->t+k and found prediction ability declined, which only shows temporal proximity is important, but does not explain why t+1. I believe bidirectional instantaneous mutual information at time t might be a better measure of synchrony. To sum up, from a multi-turn interaction perspective, t->t+1 might also be acceptable but needs justification.

R^2 as a synchrony measure may also not be appropriate, as it only measures linear correlation. For nonlinear dynamics likely present in social interaction, R^2 may not be the best criterion (though I acknowledge this is acceptable given current progress, just that other metrics for measuring nonlinear dynamics might be better).

W2. Social performance completely relies on LLM-as-a-judge. The correlation with human evaluation in Appendix E is insufficient, and the sample size may be small (21 pairs), which weakens the reliability of the evaluation. The sample for correlation analysis in Figure 6 may be too small. I understand there are considerable costs in pairing different LLMs from different families, but the model types used in issues discussed in Section 5 may still be quite limited.

W3. The causal relationship from synchrony to social performance is unclear (line 398). Although experiments show a linear relationship exists, many potential alternative explanations exist, such as stronger models having both better representations and better social engagement ability, which jointly influence social performance. The Theory of Mind and predictive coding at line 452 are also only speculative explanations. To truly determine the relationship here, intervention experiments or identifying mediating variables for analysis are needed. This could be suggested as a direction for further research.

I am willing to raise my score if the issues are adequately addressed (especially for W1).

**Questions:**

C1. At line 205, I am unsure whether using the best-case predictions layer is an appropriate choice. Although the goal of this work is not to achieve the best prediction, might using the best layer lead to unfair comparison or introduce bias? And is it reasonable to directly set negative R^2 to 0? Sensitivity analysis could be done to test whether this practice is effective.

Minor Issues:

C2. This is just my personal feeling (not necessarily a modification requirement): "neural synchrony" of LLM representations might mislead readers into thinking this is a direct biological analogy, when it is actually only a computational analogy (representational synchrony), and the neural mechanisms of LLMs and human brains are not the same. Some analogical expressions in the paper could emphasize the distinction between these two types of analogies.

C3. The parameter scale of the open-source models used may be relatively small, which limits readers' understanding of whether the conclusions can still hold at larger scales.

---

> ### Author Response · Authors · 2025-11-21
> **Response to Reviewer tXna (Part 1/4)**
>
> We thank reviewer tXna for the insightful feedback. We address the questions and concerns below.
>
> > **W1. Concerns on affine transformation rationale, temporal correspondence, and directionality.**
>
> Thank you for the thoughtful feedback. We will address your concerns point-by-point.
>
> **On the choice of affine transformation.** We agree that assuming a linear relationship between model representations is a simplification. However, our results show that this simple assumption works well in practice. Across layer pairs of various interacting LLMs, the affine transformation consistently achieves high predictive performance and generalizes well to the test set (as examples shown in Figure 2).
> This suggests that affine transformations are able to capture meaningful information between the models’ representations, and it is surprising that such a simple transformation is able to capture the representational alignment between interacting LLMs. This strong performance of affine transformations aligns with prior works suggesting LLM representations exhibit linear geometric structure across various tasks [1,2,3].
>
> To further examine whether additional model capacity is beneficial, we evaluated a two-layer MLP with a hidden dimension of 512 and ReLU activation. For training, we use AdamW with a learning rate of 1e-4, a weight decay of 0.1, and train for 200 epochs. Across three representative model pairs from each family type, we find that the nonlinear transformation yields only marginal improvements or worse generalization performance, as shown in Rebuttal Table 1.
>
> These results suggest that increasing expressiveness does not necessarily improve generalization. This supports the use of affine transformations as a simple, stable, and effective choice for measuring neural synchrony between socially interacting LLMs.
> We have added the new discussion and results in Section 6 and Appendix I of the revised manuscript.
>
> **Clarification of the $t \to t$ correspondence.** Your intuition is correct: our experimental design uses $t \to t$ correspondence. The confusion arose from a notation inconsistency in Control Group 2. In fact, the experimental condition corresponds to $k = 0$, representing the most temporally proximal representations between the two models (as defined in Section 3.2 *Dataset Construction*). The Control Group 2 should therefore be conditioned on $k \ge 1$. We have corrected the notations in Control Group 2 description, Figure 3, Figure 4, and their captions. We appreciate the reviewer for catching this error.
>
> **Directionality of synchrony.** Our synchrony measure is not unidirectional. Specifically, $SyncR^2(A, B)$ is computed as the average of $SyncR^2(A \to B)$ and $SyncR^2(B \to A)$, as described in Section 3.2 *Measuring Neural Synchrony*. This bidirectional formulation ensures that both models must be mutually predictable for a high synchrony score, avoiding the asymmetric case noted by the reviewer. We thank the reviewer for the opportunity to clarify this point, and we have revised Section 3.2 to highlight this bidirectional measure more clearly.
>
> **Rebuttal Table 1**: Comparison of nonlinear and affine transformations in measuring neural synchrony.
> | **Model Pair**                                   | **Affine**        | **Nonlinear**    |
> |--------------------------------------------------|-------------------|------------------|
> | Mistral-7B-Instruct-v0.2 & Mistral-7B-Instruct-v0.2 | 0.46 ± 0.01       | 0.47 ± 0.01      |
> | Mistral-7B-Instruct-v0.2 & Llama-3-8B-Instruct       | 0.55 ± 0.02       | 0.60 ± 0.01      |
> | Llama-3-8B-Instruct & Llama-3-8B-Instruct            | 0.75 ± 0.01       | 0.59 ± 0.01      |

---

> ### Author Response · Authors · 2025-11-21
> **Response to Reviewer tXna (Part 2/4)**
>
> > **W2 & Q3. Use of LLM-as-a-judge for social performance evaluation; sample size concerns for Figure 6; limited model types.**
>
> We appreciate the reviewer’s thoughtful concerns.
>
> **Reliability of LLM-as-a-judge.** The correlation between gpt-oss-120b and human judgments reported in Appendix E (Appendix H in the revised manuscript) follows the same human–model comparison protocol as the original Sotopia paper, with a human-annotated sample size of 180 interactions [4]. The resulting correlations are strong with over 75% of scores fall within one standard deviation of human ratings. These results indicate that gpt-oss-120b is a reliable and cost-effective evaluator for large-scale experiments, which allows us to avoid subsampling and instead evaluate all simulated interactions.
>
> **Sample size of model pairs.**
> Our correlation analysis in Figure 6 is based on *21 distinct model pairs*, which is already a relatively large sample number compared with prior cross-model representational alignment studies [5,6]. More importantly, the correlations are strong ($r = 0.88–0.99$) and statistically significant ($p < 0.05$) across all three model families, indicating that the effect is robust rather than sensitive to sample size.
>
> **Model scale and diversity.** While we focus on *Mistral*, *Llama*, and *Cross-model* family, the selection of models span multiple versions and parameter sizes, providing substantial architectural and representational variation for studying neural synchrony. Simulating and evaluating every possible pair across these models with multiple seeds requires tens of thousands of social simulations, which is computationally demanding.
> We view our results as an important initial step, presenting meaningful findings from a neuroscience perspective, and providing a foundation for future work that can expand to larger models and broader model families.
>
> > **W3. Unclear causal relationship between synchrony and social performance; alternative explanations.**
>
> We thank the reviewer for raising this important point. In the paper, we were careful not to claim a causal relationship between neural synchrony and social performance. In Figure 6, we visualize only the linear relationship without assuming any causal direction. Our intention was to highlight the empirical correlation as a fresh and interesting observation, rather than to prove causality. We do agree that establishing causality would require intervention-based experiments, and we have listed this as an important direction for future work.
>
> We agree that stronger models may jointly exhibit richer representations and higher social performance. To directly assess this possibility, we perform *partial correlation analysis* in Section 5.3 that control for two key abilities required in Sotopia, instruction following and long-context reasoning. Even after conditioning on them, the correlation between neural synchrony and social performance remains significant across all three family types (Table 1). This indicates that the observed relationship cannot be solely explained by "stronger models having both better representations and better social ability".

---

> ### Author Response · Authors · 2025-11-21
> **Response to Reviewer tXna (Part 3/4)**
>
> > **Q1. The choice of best-case prediction layers and setting negative $R^2$ to 0 when computing neural synchrony.**
>
> **Layer choice in computing $SyncR^2$.** We use the best predictive match for each layer in model A when computing $SyncR^2$, as described in Section 3. This choice is motivated by findings in neuroscience showing that inter-brain synchrony arises between specific brain regions rather than uniformly across the entire brain [7,8]. By analogy, we view these best predictive layer pairs as the most informative regions between two LLMs for measuring neural synchrony, whereas layer pairs with lower or negative test-set $R^2$ are interpreted as exhibiting no synchrony.
>
> To examine whether our conclusions depend on taking the average of only the best-matching layers, we compute $SyncR^2$ using the average of the top-$k$ predictive matches for each layer (with $k = 1$ corresponding to our original method). All other parts of the measurement remain unchanged. We then analyze how the correlation between $SyncR^2$ and social performance varies with different $k$s.
>
> As shown in Figure 7 in the revised manuscript, for the *Mistral* family, increasing $k$ strengthens the correlation between $SyncR^2$ and social performance. For the *Llama* and *Cross*-family model pairs, larger $k$ values slightly weaken this correlation. In all cases the correlation remains high ($r > 0.7, p<0.05$).
> These results indicate that our original layer selection design is a reasonable choice, but not the only one: alternative selections lead to similarly strong correlations, showing that the metric and finding are robust across different design choices.
>
> **Clipping of negative $R^2$ values.**
> A negative $R^2$ value indicates that the learned transformation fails to generalize on the test set. Formally, for each sample $i$ and output dimension $j$, $R\_j^2 = 1 - \\frac{\\sum\_i{(y\_{ij} - \\hat{y}\_{ij})^2}}{\\sum\_i{(y\_{ij}- \\bar{y}\_j)^2}}$, and $R^2=\\frac{1}{d}\\sum^d\_{j=1}{R^2\_j}$. A constant mean predictor $\hat{y_{ij}}=\bar{y_j}$ achieves $R^2_j=0$.
> $R\^2<0$ indicates that learned transformation performs worse than predicting simply the mean.
>
> To empirically assess the impact of setting negative $R^2$ values to zero, we compute $SyncR^2$ under two conditions: (1) using the original procedure where negative $R^2$ values are set to zero, and (2) keeping negative $R^2$ values. As shown in Figure 8 in the revised manuscript, keeping negative $R^2$ values substantially reduces $SyncR^2$ and leads to markedly inflated standard errors of $SyncR^2$.
> This behavior is driven by the large variance introduced by negative $R^2$ values, which arise when the predictive transformation fails to generalize better than predicting the mean.
> These results indicate that keeping negative $R^2$ leads to unstable and unreliable synchrony measurements.
>
> > **Q2. "Neural synchrony" might be misleading.**
>
> Thank you for your thoughtful comment. We would like to clarify that in the introduction (Line 46), we define "neural synchrony" specifically as the alignment between representations of interacting LLMs. We also make a distinction between biological activities and computational representations clear in the Ethics and Risks Statement.
>
> The term "neural" is used in line with common ML terminology (e.g., "neural networks," "neurons," "activations," etc.), where it refers to the structure and functioning of artificial neural networks rather than biological neurons.
>
> We appreciate your suggestion and have added a clearer clarification in the introduction (Line 44-45).

---

> ### Author Response · Authors · 2025-11-21
> **Response to Reviewer tXna (Part 4/4)**
>
> **References**
>
> [1] Park, Kiho, et al. "The Linear Representation Hypothesis and the Geometry of Large Language Models." International Conference on Machine Learning. PMLR, 2024.
>
> [2] Li, Kenneth, et al. "Inference-time intervention: Eliciting truthful answers from a language model." Advances in Neural Information Processing Systems 36 (2023): 41451-41530.
>
> [3] Kim, Junsol, et al. "Linear Representations of Political Perspective Emerge in Large Language Models." The Thirteenth International Conference on Learning Representations.
>
> [4] Zhou, Xuhui, et al. "SOTOPIA: Interactive Evaluation for Social Intelligence in Language Agents." The Twelfth International Conference on Learning Representations.
>
> [5] Huh, Minyoung, et al. "The platonic representation hypothesis." International Conference on Machine Learning. PMLR, 2024.
>
> [6] Yu, Sihyun, et al. "Representation Alignment for Generation: Training Diffusion Transformers Is Easier Than You Think." The Thirteenth International Conference on Learning Representations.
>
> [7] Dumas, Guillaume, et al. "Inter-brain synchronization during social interaction." PloS one 5.8 (2010): e12166.
>
> [8] Kawasaki, Masahiro, et al. "Inter-brain synchronization during coordination of speech rhythm in human-to-human social interaction." Scientific reports 3.1 (2013): 1692.

---

> > ### Comment · Reviewer_tXna · 2025-11-26
> > **Official Comment by Reviewer tXna**
> >
> > I appreciate the work done by the authors during the rebuttal. The additional clarifications resolved my questions. I would like to raise my score to 6.

---

> > > ### Author Response · Authors · 2025-11-26
> > > **Thank you for your feedback**
> > >
> > > Dear Reviewer tXna,
> > >
> > > Thank you for your thoughtful questions, which helped us improve the quality of the paper. We are glad that your concerns have been resolved, and we truly appreciate the time and effort you devoted to reviewing our work!

---

### Author Response · Authors · 2025-11-21
**General Response**

We appreciate the insightful and constructive feedback from the reviewers.

We are encouraged by their recognition that our paper is clear, well-written, and accessible to readers without a neuroscience background (Reviewer tXna, m5ti), offers novel insights into understanding LLM social capabilities from a neuroscience perspective (Reviewer tXna, m5ti, xLjv), provides a new tool for probing interacting LLMs (Reviewer m5ti), uncovers meaningful empirical findings (Reviewer tXna), and is supported by rigorous and well-designed experiments (Reviewer tXna, m5ti, xLjv).

In response to the reviewers’ comments, we have made the following revisions to the manuscript, which are highlighted in the current version:

- Added discussions and experiments regarding the choice of affine transformation instead of a nonlinear transformation. (Reviewer xLjv, tXna)

- Corrected the lag-$k$ notations in Control Group 2 and relevant figures. (Reviewer tXna)

- Emphasized the bidirectional measure of neural synchrony (Reviewer tXna)

- Conducted a sensitivity analysis on the choice of best-case prediction layers and setting negative $R^2$ to 0 when computing neural synchrony. (Reviewer tXna)

- Provided clarification on sampling train-test split and included additional experiments on testing whether neural synchrony is driven due to the shared persona pairs. (Reviewer m5ti)

- Expanded the discussion on future directions, including whether a causal relationship might exist and how more complex and realistic settings could be adopted. (Reviewer tXna, m5ti, xLjv)

We provide more details in the direct responses to the reviewers.

---

### Author Response · Authors · 2025-11-30
**Summary of the Discussion**

Dear Area Chair and Reviewers,

Due to the recent OpenReview incident and the reassignment of all papers to new Area Chairs, we would like to provide a brief summary of the discussion period.

Across the reviews, our work received consistently positive recognition for several key strengths:

- **Novel perspective and conceptual framing**: Reviewers highlighted that this is the first work to study representation-level neural synchrony between socially interacting LLMs, and that the neuroscience-inspired framing provides a fresh and well-motivated tool for probing LLM behavior beyond surface outputs (Reviewer tXna S1, Reviewer m5ti S1, Reviewer xLjv S1)

- **Thorough and rigorous experimental design**: Reviewers emphasized the careful controls, systematic analyses addressing potential confounding factors, and broad experimental coverage. (Reviewer tXna S2, Reviewer m5ti S2, Reviewer xLjv S2)

- **Meaningful empirical findings**: Reviewers recognized the practically meaningful findings that representation-level neural synchrony and behavioral-level social performance are strongly correlated (Reviewer tXna S3).

- **Clarity and accessibility**: The paper was commented as clearly written, well-structured, and highly accessible even to readers without a neuroscience background (Reviewer tXna S2, Reviewer m5ti S3).

During the discussion period, in response to the reviewers’ thoughtful feedback, we addressed all major concerns with detailed clarifications, additional experiments, and further analyses. We have incorporated these revisions into the updated manuscript, which further improved the quality of our work. A detailed revision summary is provided in the "General Response" comment.

After the response, Reviewer tXna confirmed that their concerns were resolved and raised their score accordingly. All reviewers remained positive, and we believe that no substantive concerns remain open.

We sincerely thank the reviewers again for their insightful and constructive feedback, and we also appreciate in advance the Area Chair’s valuable time and consideration in evaluating our submission.

---

### Meta-Review · Area_Chair_R4fe · 2026-01-08

**Summary:**

This paper makes a novel and timely contribution by introducing a neuroscience-inspired perspective on social interaction in LLMs. Reviewers consistently praised the rigorous and carefully controlled experimental design, the breadth of evaluations across multiple model pairs, and the clear, accessible presentation. The main weaknesses identified concern theoretical justification of the affine mapping and synchrony metric, limited causal claims, and questions about scale and controls. I believe these concerns were framed as addressable rather than fundamental. Most of them have been addressed during the rebuttal phase. Therefore, I recommend for acceptance.

**Reviewer Concerns:**

Justification of affine mapping and synchrony definition (tXna/xLjv)
Controls for shared context, persona leakage, and confounds (m5ti)
Evaluation validity and causality limits (tXna)

**Reviewer Scores:**

Reviewer tXna will raise the scores.

---

### Decision · Program_Chairs · 2026-01-26

Accept (Poster)